# Minerva: Reinforcement Learning with Verifiable Rewards for Cyber Threat Intelligence LLMs

## Abstract

Cyber threat intelligence (CTI) analysts routinely convert noisy, unstructured security artifacts into standardized, automation-ready representations. Although large language models (LLMs) show promise for this task, existing approaches remain brittle when producing structured CTI outputs and have largely relied on supervised fine-tuning (SFT). In contrast, CTI standards and community-maintained resources define canonical identifiers and schemas that enable deterministic verification of model outputs. We leverage this structure to study reinforcement learning with verifiable rewards (RLVR) for CTI tasks. We introduce *Minerva*, a unified dataset and training pipeline spanning multiple CTI subtasks, each paired with task-specific verifiers that score structured outputs and identifier predictions. To address reward sparsity during rollout, we propose a lightweight self-training mechanism that generates additional verified trajectories and distills them back into the model. Experiments across LLM backbones show consistent improvements in accuracy and robustness over SFT across multiple benchmarks.

## 1. Introduction

Cyber threat intelligence (CTI) supports modern defensive security workflows by providing shared, machine-readable representations for triage, detection engineering, and incident response (Xu et al., 2024). In practice, analysts map heterogeneous, unstructured artifacts, such as vulnerability descriptions and incident narratives, into actionable intelligence grounded in standardized frameworks and identifiers, including MITRE ATT&CK for adversary behavior and widely used vulnerability standards for identification and severity scoring (MITRE Corporation, 2026d; Byers

et al., 2022; MITRE Corporation, 2026c; FIRST.org, Inc., 2019). These standards enable consistent reporting and large-scale exchange across organizations and tools, including Structured Threat Information Expression (STIX) and Trusted Automated Exchange of Indicator Information (TAXII) (Strom et al., 2020; OASIS Cyber Threat Intelligence (CTI) Technical Committee, 2021a;b). However, they also raise the bar for correctness: CTI pipelines require accurate grounding to evolving identifiers, faithful extraction from noisy text, and precise mapping to canonical concepts. As CTI volume grows, this expert mapping effort remains a bottleneck, and even small factual or formatting errors can silently break automation and propagate downstream.

LLMs are well suited to this setting because they can ingest long-form narratives and produce structured outputs for downstream workflows. Surveys document increasing LLM use across cybersecurity tasks, and domain-adapted models such as CTI-BERT show that security-specific pretraining improves CTI-oriented extraction and representations (Xu et al., 2024; Park & You, 2023). Yet benchmarks such as SEvenLLM, CTIBench, and CyberBench report uneven reliability: models often follow analyst-style instructions and recover surface facts, but still miss workflow-critical outputs, including ATT&CK mapping, mitigation recommendation, and vulnerability root-cause identification (Ji et al., 2024; Alam et al., 2024; Liu et al., 2024). These failures reflect dense domain terminology, the need for syntactically valid, semantically grounded, structured outputs, and deployment constraints that favor smaller, specialized models. CTI also offers many verifiable targets, enabling explicit rewards and verifier-driven learning beyond imitation, motivating *reinforcement learning with verifiable rewards* (RLVR), a scalable training paradigm for CTI expert models.

RLVR replaces learned preference models with deterministic, programmatic verifiers that score a completion by checking structured properties, such as identifier correctness and output format. Recent work shows that RLVR can substantially improve reasoning and structured generation in LLMs (Shao et al., 2024; DeepSeek-AI et al., 2025; Wen et al., 2025), while avoiding the cost and subjectivity of human preference collection in RLHF-style pipelines (Ouyang et al., 2022). However, standard on-policy RLVR

[1]Anonymous Institution, Anonymous City, Anonymous Region, Anonymous Country. Correspondence to: Anonymous Author <anon.email@domain.com>.

Preliminary work. Under review by the International Conference on Machine Learning (ICML). Do not distribute.

can be constrained by the base model's empirical support: under small rollout budgets, some prompts yield no verified-correct samples, providing little learning signal in that iteration (Wu et al., 2025). We observe this reward sparsity regime frequently in CTI tasks that require precise grounding of long-tail identifiers and strict schemas.

In this work, we introduce **Minerva**, a CTI expert LLM trained on a large curated dataset spanning three modalities: (i) vulnerability-centric mapping (e.g., CVE → CWE/CVSS/ATT&CK), (ii) detection-centric mapping (e.g., Sigma/Sentinel/Splunk → ATT&CK), and (iii) procedure and attribution-oriented mapping (e.g., behaviors or scenarios → techniques, mitigations, or actors). All tasks share a structured target space with canonical identifiers and deterministic verification. To train reliably under reward sparsity, we propose **MinervaRL**. This RLVR extension augments hard prompts with *answer-conditioned* generation, providing the correct label together with a canonical reference for that label to support grounded reasoning and increase the chance of producing verified trajectories. We then distill these verified trajectories back to the original, answer-free prompts via a lightweight supervised step, improving learning on prompts that would otherwise contribute zero reward under limited sampling. Our key contributions are as follows.

- We curate **Minerva-CTI**, a unified 16-task CTI training suite with deterministic, verifier-checkable targets spanning vulnerability, detection, and procedure-oriented CTI workflows.

- We propose **MinervaRL**, an RLVR extension that addresses reward sparsity via hardness-gated answer-conditioned reasoning and periodic distillation back to the answer-free prompts.

- We demonstrate consistent gains on 12 CTI benchmarks across four open-weight backbones, with MinervaRL outperforming strong instruction-tuned and security SFT baselines.

## 2. Related Work

**LLMs for CTI.** A central CTI challenge is extracting tactics, techniques, and procedures (TTPs) from natural-language threat reports and grounding them to ATT&CK. TRAM demonstrates an applied LLM pipeline for automated technique identification, motivated by the cost and brittleness of manual mapping (Center for Threat-Informed Defense, 2023). A recent systematization survey automated TTP extraction methods—including generative LLMs—and notes persistent comparability issues due to heterogeneous ontologies, datasets, and evaluation protocols (Büchel et al., 2025). To support more controlled training and evalua-

tion, AnnoCTR releases expert-annotated CTI corpora labeled with ATT&CK techniques (Lange et al., 2024). Beyond report text, LLMs have been used to bootstrap structured CTI artifacts such as knowledge graphs (Hu et al., 2024) and to map vulnerability descriptions into standardized taxonomies, where prompting alone remains unreliable. However, instruction templating and fine-tuning (e.g., VTT-LLM) can improve grounding (Liu et al., 2023; Zhang et al., 2024). Instruction-tuning efforts such as Cyber-Pal/CyberPal 2.0 show further gains while underscoring the limitations of supervised fine-tuning for robust CTI behavior (Levi et al., 2024; 2025). Complementary directions include data-efficient technique identification (e.g., active learning) (Rahman et al., 2024) and pipelines that map detection artifacts (e.g., Sigma rules, SIEM analytics) to ATT&CK via prompt chaining and retrieval (Wudali et al., 2025).

**CTI benchmarks for LLM evaluation.** A growing set of benchmarks evaluates LLMs on core cyber threat intelligence (CTI) workflows and exposes domain-specific failure modes such as hallucination, mis-grounding, and brittle identifier mapping. CTIBench and its successor AthenaBench cover tasks including CVE→CWE mapping, CVSS scoring, ATT&CK technique extraction, and threat-actor attribution, providing targeted CTI assessment beyond general NLP benchmarks (Alam et al., 2024; 2025). SEvenLLM contributes a bilingual instruction corpus and benchmark oriented toward incident analysis and response-style CTI tasks (Ji et al., 2024). CyberMetric offers 10,000 questions to probe broad cybersecurity knowledge in LLMs (Tihanyi et al., 2024), while CyberBench aggregates 10 datasets spanning entity recognition, summarization, and classification for cybersecurity-language understanding (Liu et al., 2024).

**RLVR for LLMs.** Reinforcement learning with verifiable rewards (RLVR) has become a standard approach for improving LLM reasoning in settings where correctness can be checked programmatically, most prominently in math and programming (Shao et al., 2024; DeepSeek-AI et al., 2025). At the same time, recent work debates whether RLVR can elicit genuinely new reasoning capability beyond what is already latent in the base model (Yue et al., 2025; Cheng et al., 2025; Wu et al., 2025). Empirical evidence nonetheless suggests that RLVR can repair step-by-step reasoning behaviors and improve reliability and calibration relative to supervised baselines (Wen et al., 2025). Follow-up studies examine how gains depend on task difficulty and the exploration scale, underscoring the need to balance reasoning depth and breadth during training (Yang et al., 2025). Beyond in-distribution improvements, RLVR has been shown to enhance out-of-distribution generalization on causal reasoning tasks (Lu et al., 2025) and to transfer beyond math/code by enabling verifiable-reward formulations across diverse domains while preserving robustness and reliability gains

(Su et al., 2025).

## 3. Minerva-CTI Dataset

We introduce Minerva-CTI, a unified cyber threat intelligence (CTI) dataset curated from a range of standards and community resources. These sources include MITRE ATT&CK (MITRE Corporation, 2026d;a), MITRE CAPEC, the National Vulnerability Database (NVD) (Byers et al., 2022; MITRE Corporation, 2026c), MITRE Mapping Explorer (Center for Threat-Informed Defense, 2026), and detection and emulation corpora such as Sigma, Atomic Red Team, Microsoft Sentinel, and Splunk Security Content (SigmaHQ, 2026; Red Canary, 2026; Microsoft, 2026; Splunk, 2026). From these resources, we define 16 tasks, each characterized by a specific input and a verifiable output, such as canonical identifiers or structured label sets. The tasks encompass predicting ATT&CK techniques, tactics, and mitigations from natural-language excerpts; identifying Common Weakness Enumeration (CWE) entries and predicting vulnerability severity from vulnerability descriptions; attributing threat actors to observed behaviors; and predicting ATT&CK techniques or tactics from detection rules. The dataset comprises 32,000 training and 1,200 validation instances. Appendix A provides further details on dataset sources and splits.

## 4. Methodology

### 4.1. Reinforcement Learning with Verifiable Rewards

We train models with reinforcement learning with verifiable rewards (RLVR), where each task is paired with a deterministic, programmatic verifier that scores a generated output $y$ for a prompt $x$. The verifier produces a scalar reward $r(x,y) \in [0,1]$ via exact matching (e.g., canonical IDs), structured partial credit (e.g., technique vs. sub-technique), or set-based overlap (e.g., tactics/mitigation/CWE sets). This setting is a natural fit for CTI because many targets are structured and automatically auditable, enabling scalable optimization without a learned reward model or human-labeled preferences. Appendix B details answer extraction and task-specific verification. We do not use any additional format reward in our experiments.

We optimize the policy $\pi_\theta$ using Group Relative Policy Optimization (GRPO) (Shao et al., 2024; DeepSeek-AI et al., 2025), a PPO-style method (Schulman et al., 2017) that avoids training a critic by estimating advantages from a group of samples per prompt. Concretely, for each prompt $x$ we sample a group of completions from $\pi_{\theta_{old}}(\cdot \mid x)$, score them with the verifier, and compute relative (group-normalized) advantages to form the GRPO policy-gradient update; we follow the standard GRPO formulation in prior work. Verifiable-reward training has recently shown

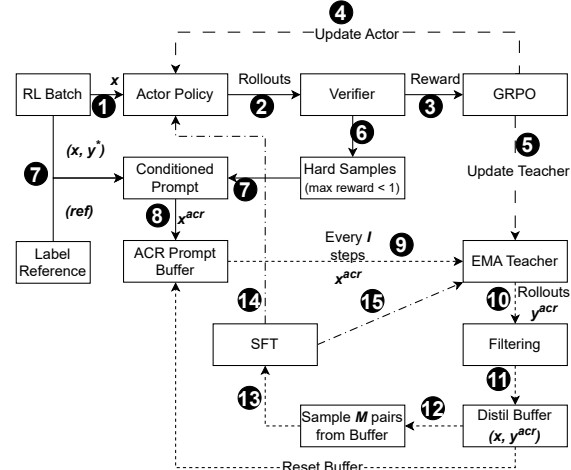

*Figure 1.* MinervaRL overview.

strong gains for structured reasoning and generation in LLMs (Shao et al., 2024; DeepSeek-AI et al., 2025; Wen et al., 2025), and GRPO provides a stable, critic-free optimizer for leveraging these signals at scale.

### 4.2. MinervaRL

Minerva-CTI tasks have *finite* answer spaces (IDs or small sets of IDs), but the effective label space can be large and long-tailed. In our snapshots, for example, the CWE taxonomy contains hundreds of distinct IDs (e.g., 944 in our January 2026 snapshot), and some standards (e.g., MITRE Enterprise Mitigations; 44 mitigations in our snapshot) are substantially less prevalent in pretraining data than popular schemas such as ATT&CK techniques or CWEs. As a result, under RL with verifier rewards (RLVR), many prompts yield *zero* successful rollouts in a batch. When no rollout receives a reward, the update becomes effectively signal-free, exacerbating reward sparsity and slowing learning on these hard, low-coverage labels.

MinervaRL is a self-training augmentation to the standard GRPO-based RLVR loop that targets hard examples without requiring human-written rationales or distillation from a larger model. The key idea is to use the ground-truth label during training to elicit a short, well-formed explanation trace (**answer-conditioned reasoning**, ACR), then distill accepted traces back into the policy using the *original* task prompt (with no label hints). Thus, inference-time prompts never expose ground-truth labels, while the policy learns to produce correct, verifiable outputs with coherent justifications.

**Notation.** Let $\mathcal{D} = \{(x_i, y_i^\star)\}$ be the training set, where $x_i$ is the original Minerva prompt and $y_i^\star$ is the ground-truth

label (ID, set of IDs, or task-specific structured label). The actor policy is $\pi_\theta$ and the EMA teacher is $\pi_\phi$. Rewards are computed by the Minerva verifier $R_{\text{minerva}}(x, y, y^\star) \in [0, 1]$, which extracts the final predicted label from a completion and checks it against $y^\star$.

**Step 1 (Algorithm 1): RLVR rollouts + GRPO update.** At each training step $t$, sample a batch $\mathcal{B}_t \subset \mathcal{D}$. For each $(x_i, y_i^\star) \in \mathcal{B}_t$, draw $N = 8$ rollouts from the actor:

$$y_{i,j}^{\text{rlvr}} \sim \pi_{\theta_t}(\cdot \mid x_i), \quad j = 1, \ldots, N, \tag{1}$$

score each rollout with the verifier,

$$r_{i,j}^{\text{base}} = R_{\text{minerva}}(x_i, y_{i,j}^{\text{rlvr}}, y_i^\star), \tag{2}$$

and update the actor with GRPO on these on-policy trajectories.

**Step 2 (Algorithm 1): Buffer hard examples for ACR.** For each prompt, compute the per-example maximum reward $m_i = \max_j r_{i,j}^{\text{base}}$. If $m_i < 1$ (and the task is not CVSS), we construct an answer-conditioned prompt

$$x_i^{\text{acr}} = x_i \oplus b(y_i^\star, d_i), \tag{3}$$

where $b(\cdot)$ is a fixed template that reveals the gold label $y_i^\star$ and requests a short justification, and $d_i$ optionally contains truncated canonical label details. We add $x_i^{\text{acr}}$ to an ACR prompt buffer $\mathcal{P}$ for deferred generation.

**Step 3 (Algorithm 1): Deferred ACR generation + filtering.** Every $I = 10$ training steps, we sample $K = 4$ ACR completions for each buffered prompt using an EMA teacher $\pi_{\phi_t}$ (temperature $\tau = 0.7$, nucleus $p = 0.9$):

$$y_{i,k}^{\text{acr}} \sim \pi_{\phi_t}(\cdot \mid x_i^{\text{acr}}), \quad k = 1, \ldots, K. \tag{4}$$

Each candidate is scored by the verifier, then passed through the two-stage filtering pipeline in Appendix C (heuristics + TextCNN ML filter). The ML filter outputs a GOOD probability $q_{i,k} \in [0, 1]$, and we only retain candidates with $q_{i,k} \geq \tau_q$ (we use $\tau_q = 0.5$). If any candidates remain, we select $k^\star = \arg\max_{k \in \mathcal{E}_i} q_{i,k}$ and enqueue the selected distillation pair $(x_i, y_{i,k^\star}^{\text{acr}})$ into a distillation buffer $\mathcal{Q}$.

**Step 4 (Algorithm 1): SFT distillation.** At each distillation interval, we sample up to $M = 256$ pairs from $\mathcal{Q}$ and perform one SFT update on the actor with learning rate $\gamma \cdot \text{lr}_{\text{rlvr}}$ (we use $\gamma = 0.05$), then flush buffers $\mathcal{P}$ and $\mathcal{Q}$.

### 4.3. Expanding Empirical Support with MinervaRL

Under a fixed rollout budget $k$ per prompt, on-policy RLVR can stall on "zero-reward" prompts simply because verified-correct outputs are too rare to be observed. Let $p_\theta(a^\star \mid x)$

---

**Algorithm 1:** MinervaRL

**Input:** Dataset $\mathcal{D}$; actor $\pi_\theta$; verifier $R_{\text{minerva}}$; RL rollouts $N$; ACR rollouts $K$; distill interval $I$; EMA decay $\alpha$; distill batch cap $M$; RLVR learning rate $\text{lr}_{\text{rlvr}}$; lr scale $\gamma$

**Initialize:** EMA teacher $\phi \leftarrow \theta$; ACR buffer $\mathcal{P} \leftarrow \emptyset$; distill buffer $\mathcal{Q} \leftarrow \emptyset$;

**for** $t = 1, 2, \ldots$ **do**

  **(1) RLVR rollouts + GRPO update**;
  Sample RL batch $\mathcal{B}_t \subset \mathcal{D}$;
  **foreach** $(x_i, y_i^\star) \in \mathcal{B}_t$ **do**
    Sample RLVR rollouts $\{y_{i,j}^{\text{rlvr}}\}_{j=1}^N \sim \pi_\theta(\cdot \mid x_i)$;
    $r_{i,j}^{\text{base}} \leftarrow R_{\text{minerva}}(x_i, y_{i,j}^{\text{rlvr}}, y_i^\star)$;
    $m_i \leftarrow \max_j r_{i,j}^{\text{base}}$;
    **if** $m_i < 1 \wedge \text{task}(x_i) \neq \text{CVSS}$ **then**
      **(2) Buffer hard examples for ACR**;
      Build $x_i^{\text{acr}} \leftarrow x_i \oplus b(y_i^\star, d_i)$; add $x_i^{\text{acr}}$ to $\mathcal{P}$;

  Update actor with GRPO: $\theta \leftarrow \text{GRPO}(\theta, \mathcal{B}_t)$;
  Update EMA teacher: $\phi \leftarrow \alpha\phi + (1 - \alpha)\theta$;
  **if** $t \bmod I = 0$ **then**
    **(3) Deferred ACR generation + filtering**;
    **foreach** $x_i^{\text{acr}} \in \mathcal{P}$ **do**
      Sample $\{y_{i,k}^{\text{acr}}\}_{k=1}^K \sim \pi_\phi(\cdot \mid x_i^{\text{acr}})$;
      $\mathcal{E}_i \leftarrow \{k \in \{1, \ldots, K\} \mid$
      $R_{\text{minerva}}(x_i, y_{i,k}^{\text{acr}}, y_i^\star) =$
      $1 \wedge f_{\text{heur}}(y_{i,k}^{\text{acr}}) = 1 \wedge q_{i,k} \geq \tau_q\}$;
      **if** $\mathcal{E}_i \neq \emptyset$ **then**
        $k^\star \leftarrow \arg\max_{k \in \mathcal{E}_i} q_{i,k}$;
        Append $(x_i, y_{i,k^\star}^{\text{acr}})$ to $\mathcal{Q}$;

    **(4) SFT distillation**;
    Sample $\mathcal{S} \subseteq \mathcal{Q}$ with $|\mathcal{S}| \leq M$; do one SFT step with lr $\gamma \cdot \text{lr}_{\text{rlvr}}$;
    Flush buffers $\mathcal{P} \leftarrow \emptyset$ and $\mathcal{Q} \leftarrow \emptyset$;

---

be the probability that a completion sampled from $\pi_\theta(\cdot \mid x)$ extracts the correct structured answer $a^\star$. Then the probability of observing *no* success in $k$ independent rollouts is

$$\Pr[\text{no success in } k \text{ rollouts}] = (1 - p_\theta(a^\star \mid x))^k.$$

Equivalently, if we define a detectability threshold $\varepsilon_{k,\zeta} := \frac{-\log \zeta}{k}$, then whenever $p_\theta(a^\star \mid x) \ll \varepsilon_{k,\zeta}$, we expect many iterations where all $k$ rollouts receive zero reward, yielding little or no learning signal for that prompt.

MinervaRL mitigates this sampling barrier by (i) using the gold label during training to form an answer-conditioned prompt that makes at least one verified-correct trace more likely to appear, and (ii) distilling the accepted trace back

onto the *original* prompt (without label hints), thereby increasing $p_\theta(a^\star \mid x)$ so that future RLVR rollouts on $x$ become empirically detectable under the same budget $k$.

A detailed formalization (assumptions, theorem statement, and proof sketch) is provided in Section F.

# 5. Experiments

## 5.1. Evaluation Benchmarks

We report results on the 12 evaluation tasks shown in Table 1, spanning multiple-choice QA, structured CTI taxonomy prediction, SOC-style reasoning, and information extraction. Full task definitions, output formats, and sample counts are provided in Appendix E.

**Tasks.** **CKT(Alam et al., 2025):** CTI multiple-choice questions with five options (A–E). **CyberMetric(Tihanyi et al., 2024):** cybersecurity multiple-choice questions with four options (A–D). **SOCEval(Deason et al., 2025):** Security Operations Center (SOC)-style multi-select QA over threat intel reports. **RCM (Alam et al., 2025):** map a CVE description to its underlying CWE. **VSP (Alam et al., 2025):** produce a CVSS v3.1 base vector string. **ATE (Alam et al., 2025):** identify the most relevant MITRE ATT&CK technique for a described attack scenario. **RMS (Alam et al., 2025):** recommend ATT&CK mitigations for a scenario. **ElasticRule:** map an Elastic detection rule to a single MITRE ATT&CK technique ID. **APTNER (Wang et al., 2022):** APT-focused named-entity recognition. **LANCE(Froudakis et al., 2025):** indicator of compromise (IoC) identification over candidate indicators (aggregated over IP/URL/Domain/Hash subtasks). **AnnoCTR(Lange et al., 2024):** Structured Threat Information Expression (STIX)-style entity/relation extraction (meta task aggregated over four subtasks). **AZERG(Lekssays et al., 2025):** STIX-style entity/relation extraction (meta task aggregated over four subtasks).

**Contamination and overlap considerations.** Fully eliminating train–test overlap is difficult for CTI benchmarks, so we design Minerva-CTI to minimize leakage where possible. We exclude multiple-choice formats during training to reduce direct overlap with MCQ knowledge benchmarks such as CKT and CyberMetric. SOCEval is derived from threat-intel reports that are not used as training targets. For CVE-based tasks (RCM, VSP), both training and evaluation draw on NVD-style descriptions; to reduce temporal leakage, we restrict Minerva-CTI to pre-2025 CVEs, while AthenaBench evaluations emphasize 2025-era entries. For ATT&CK scenario tasks (ATE, RMS), Minerva-CTI uses ATT&CK-derived scenarios, whereas evaluation uses independently curated (model-generated) scenarios conditioned on technique descriptions. Finally, ElasticRule, APTNER,

LANCE, AnnoCTR, and AZERG are included only for evaluation and are not used as supervised training objectives.

## 5.2. Experimental Settings

**RLVR (GRPO) training.** We train with GRPO using a per-step training batch size of 128 prompts. For each prompt, we sample $N = 8$ on-policy rollouts, truncate prompts to 2048 tokens and responses to 1024 tokens, and optimize the actor with learning rate $1 \times 10^{-6}$ for 500 steps. For checkpoint selection, we monitor a combined validation set consisting of Minerva-Dev and the AthenaBench-Mini subset (Alam et al., 2025), and select the best checkpoint based on the average performance across these validation splits.

**MinervaRL.** For prompts that fail to yield a verified-correct rollout, MinervaRL generates answer-conditioned reasoning (ACR) traces using an exponential moving average (EMA) teacher (decay $\alpha = 0.995$), sampling $K = 4$ traces per ACR prompt at temperature $0.7$ with nucleus sampling $p = 0.9$. ACR prompts allow a longer context window (4096 tokens) than base RLVR prompts. Every $I = 10$ steps we distill up to 256 accepted traces back onto the original (answer-free) prompts using supervised fine-tuning with learning rate scaled by $\gamma = 0.05$ relative to the RLVR learning rate. Full hyperparameter details (including filtering thresholds and other knobs) are provided in Appendix G.

**LLM Backbones.** We train Minerva on four open-weight backbones: **Llama-3.2-3B-Instruct** and **Llama-3.1-8B-Instruct** (Meta, 2024b;a), as well as **Qwen3-4B-Base** and **Qwen3-8B-Base** (Qwen Team, 2025a;b). For each backbone, we report the base model (no CTI adaptation), an RLVR model trained with GRPO ("-GRPO"), and a MinervaRL model trained with the same GRPO loop plus hardness-gated answer-conditioned reasoning and periodic distillation ("-MinervaRL").

To contextualize Llama-3.1-8B performance, we additionally include two supervised fine-tuned (SFT) security baselines built on the Llama-3.1-8B family: **Llama-Primus-Merged** and **Foundation-Sec-8B-Instruct** (Llama-Primus Team, 2025; Foundation AI, 2025).

**Evaluation Metrics.** We report the metrics defined by each benchmark/task: multiple-choice tasks use exact-match accuracy on the extracted option; taxonomy and mapping tasks (e.g., ElasticRule/ATE, RCM) use exact-match of the extracted label (MITRE technique ID or CWE); structured extraction tasks (e.g., APTNER) use micro-averaged F1 over JSON entities; RMS uses multi-label F1 over mitigation sets; and suites with multiple subtasks (e.g., LANCE, AnnoCTR, AZERG) report averages over their constituent subtasks. For VSP, we follow the benchmark verifier and

*Table 1.* Benchmark results across evaluation suites.

| Model | CKT | CyberMetric | SOCEval | RCM | VSP | ATE | RMS | ElasticRule | APTNER | LANCE | AnnoCTR | AZERG | Avg. |
|---|---|---|---|---|---|---|---|---|---|---|---|---|---|
| Llama-3.1-8B-Instruct | 67.6 | 83.2 | 64.8 | 48.4 | 76.0 | 17.4 | 6.7 | 14.4 | 33.5 | 78.2 | 50.7 | 42.8 | 48.6 |
| Llama-Primus-Merged | 76.5 | 85.9 | 68.0 | 56.0 | 72.6 | 33.6 | 7.8 | 27.3 | 22.0 | 59.4 | **51.8** | 40.9 | 50.2 |
| Foundation-Sec-8B-Instruct | 77.1 | 81.3 | 67.9 | 61.0 | 65.8 | 39.2 | 15.4 | 33.6 | 34.0 | 59.5 | 43.5 | **44.4** | 51.9 |
| Llama-3.1-8B-GRPO | 71.9 | 85.4 | 63.0 | 66.3 | 82.6 | 32.0 | 30.9 | 32.2 | 32.7 | **86.2** | 49.5 | 39.5 | 56.0 |
| Llama-3.1-8B-MinervaRL | 73.9 | 84.2 | 64.7 | **68.8** | 87.6 | **48.4** | **42.1** | **40.5** | 34.1 | 84.6 | 50.3 | 43.7 | **60.2** |
| Llama-3.2-3B-Instruct | 71.6 | 77.0 | 55.7 | 15.2 | 2.8 | 1.0 | 0.4 | 1.2 | 25.4 | 77.8 | 35.9 | 32.8 | 33.1 |
| Llama-3.2-3B-GRPO | 71.2 | 77.8 | 58.3 | 48.9 | 56.5 | 5.2 | 15.4 | 5.3 | 27.5 | 70.6 | 38.5 | 29.7 | 42.1 |
| Llama-3.2-3B-MinervaRL | 71.0 | 78.1 | 54.6 | 57.2 | 77.1 | 21.8 | 29.3 | 20.1 | 16.7 | 82.2 | 37.9 | 33.7 | 48.3 |
| Qwen3-8B-Base | 71.7 | 69.3 | 63.8 | 52.6 | 69.0 | 13.4 | 6.5 | 9.0 | 31.3 | 45.1 | 9.8 | 9.8 | 37.6 |
| Qwen3-8B-GRPO | 69.8 | 72.7 | **69.0** | 60.5 | 68.8 | 23.6 | 8.2 | 18.1 | **37.7** | 66.1 | 37.8 | 31.1 | 47.0 |
| Qwen3-8B-MinervaRL | **77.8** | **88.2** | 67.5 | 64.8 | 79.4 | 32.0 | 20.1 | 22.0 | 37.4 | 74.9 | 43.0 | 33.1 | 53.4 |
| Qwen3-4B-Base | 45.6 | 50.1 | 56.1 | 45.6 | 76.5 | 4.2 | 6.5 | 4.6 | 1.2 | 18.0 | 16.6 | 6.5 | 27.6 |
| Qwen3-4B-GRPO | 74.2 | 85.8 | 63.9 | 60.8 | **88.1** | 20.2 | 3.8 | 20.4 | 32.4 | 58.2 | 39.6 | 24.0 | 47.6 |
| Qwen3-4B-MinervaRL | 70.0 | 80.0 | 64.0 | 59.9 | 80.0 | 25.4 | 5.1 | 27.5 | 28.0 | 56.7 | 50.4 | 29.1 | 48.0 |

compute a normalized score from the CVSS v3.1 base-score mean absolute deviation (MAD), $1 - \mathrm{MAD}/7.7$.

## 6. Results & Analysis

### 6.1. Main Results

**GRPO consistently improves CTI performance.** Across all four backbones, RLVR with GRPO improves the average score in Table 1 over the base model: Llama-3.1-8B (48.6→56.0), Llama-3.2-3B (33.1→42.1), Qwen3-8B (37.6→47.0), and Qwen3-4B (27.6→47.6). This suggests verifiable rewards provide a strong learning signal for structured CTI outputs even without a learned reward model.

**MinervaRL yields further gains over GRPO.** MinervaRL improves over GRPO on every backbone, with an average relative gain of **8.9%** across the four models (Avg: 48.18→52.48). On Llama-3.1-8B, MinervaRL raises the average score from 48.6 to 60.2 (**23.9%** over base; **16.0%** over the strongest public SFT baseline on this backbone, Foundation-Sec-8B-Instruct at 51.9). On Llama-3.2-3B, it improves from 33.1 to 48.3 (**45.9%** over base; **14.7%** over GRPO). On Qwen3-8B, it increases from 37.6 to 53.4 (**42.0%** over base; **13.6%** over GRPO). On Qwen3-4B, it improves from 27.6 to 48.0 (**73.9%** over base; **0.8%** over GRPO), suggesting GRPO contributes most of the gains at this scale while MinervaRL provides a smaller additional lift.

### 6.2. In-Training vs. Not-in-Training Tasks

We split the 12 benchmarks in Table 1 into tasks that are directly seen during Minerva-CTI training (**Seen**: RCM, VSP, ATE, RMS) and tasks that are not directly optimized as training objectives (**Not-in-training**: CKT, CyberMetric, SOCEval, ElasticRule, APTNER, LANCE, AnnoCTR,

*Table 2.* Average performance on seen vs. not-in-training tasks.

| Backbone | Variant | Seen (4) | Not-in-training (8) |
|---|---|---|---|
| Llama-3.1-8B | Base | 37.1 | 54.4 |
| Llama-3.1-8B | GRPO | 53.0 | 57.5 |
| Llama-3.1-8B | MinervaRL | 61.7 | 59.5 |
| Llama-3.2-3B | Base | 4.9 | 47.2 |
| Llama-3.2-3B | GRPO | 31.5 | 47.4 |
| Llama-3.2-3B | MinervaRL | 46.4 | 49.3 |
| Qwen3-8B | Base | 35.4 | 38.7 |
| Qwen3-8B | GRPO | 40.3 | 50.3 |
| Qwen3-8B | MinervaRL | 49.1 | 55.5 |
| Qwen3-4B | Base | 33.2 | 24.8 |
| Qwen3-4B | GRPO | 43.2 | 49.8 |
| Qwen3-4B | MinervaRL | 42.6 | 50.7 |

AZERG). For each model variant we report the mean score over tasks in the group.

As expected, GRPO and MinervaRL deliver the largest gains on the *seen* tasks (which align closely with the verifiable training objectives), but we also observe consistent improvements on the *not-in-training* suite, indicating that training on verifiable CTI tasks transfers to related CTI reasoning and extraction capabilities.

### 6.3. Qualitative Analysis

**Goal.** Benchmark scores in Section 6.1 capture whether a model can produce *verifiably correct* structured outputs. However, CTI work often demands more than correctness: analysts prefer responses that are concise, well-structured, easy to audit, and actionable. We therefore complement quantitative results with a response-quality study among five Llama-3.1-8B-based variants: **Llama-3.1-8B-Instruct**, **Llama-Primus-Merged**, **Foundation-Sec-8B-Instruct**, **Llama-3.1-8B-Instruct-GRPO**, and **Llama-3.1-8B-Instruct-MinervaRL**.

### Pairwise Preference Judge Prompt

You are a cyber threat intelligence (CTI) analyst expert. You are given a question and two LLM responses. Both responses are verifiably correct. Select the response you would prefer to receive for CTI analysis.
Question: {question}
Response A: {response_a}
Response B: {response_b}
Return JSON only:
{"winner": "A"|"B"|"tie",
"rationale": "<short rationale>"}

*Figure 2.* Prompt template used for pairwise response-quality judging (both responses are verifier-correct).

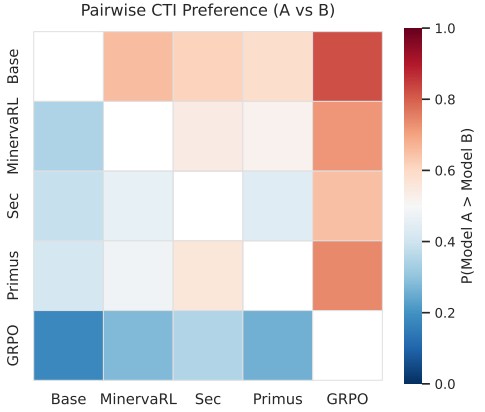

*Figure 3.* Pairwise preference heatmap from the CTI judge. Each cell (row A, column B) shows the fraction of comparisons where A is preferred over B (ties excluded). Higher values indicate stronger preference for the row model.

**Verifier-controlled evaluation set.** To avoid conflating quality with correctness, we restrict to prompts for which *all* compared models are verifier-correct. For each evaluation task we sample up to 100 such instances; in total this yields 752 prompts drawn from the benchmark datasets.

**Pairwise preference judging.** We use GPT-5.2 as an external pairwise judge to decide which of two verifier-correct responses is preferable for CTI analysis. To mitigate positional bias, we randomly shuffle which model appears as Response A vs. Response B per comparison. We intentionally avoid providing a task-specific scoring rubric beyond a single instruction to choose the preferred response, to reduce judge anchoring and reward hacking. The exact judge prompt template is shown in Figure 2.

**Results** Figure 3 summarizes the pairwise preferences among verifier-correct responses. Overall, **Llama-3.1-8B-**

*Table 3.* Performance on other benchmarks (values in %; higher is better for the first three columns, lower is better for WMDP-Cyber).

| Model | MMLU Pro ↑ | IFEval ↑ | Canary Exploit ↑ | WMDP Cyber ↓ |
|---|---|---|---|---|
| Llama-3.1-8B-Instruct | 44.1 | 72.3 | 10.9 | 46.7 |
| Llama-Primus-Merged | 41.5 | 68.2 | 29.8 | 46.3 |
| Foundation-Sec-8B-Instruct | 42.5 | 67.1 | 0.0 | 45.3 |
| Llama-3.1-8B-GRPO | 45.6 | 72.3 | 0.0 | 47.1 |
| Llama-3.1-8B-MinervaRL | 43.6 | 73.8 | 28.4 | 47.1 |

*Table 4.* Baseline comparison and component ablations using aggregate scores on Minerva-Dev and AthenaBench-Mini.

| Approach | Minerva Dev | AthenaBench Mini | Avg |
|---|---|---|---|
| Base | 18.2 | 43.0 | 30.6 |
| GRPO | 50.3 | 57.4 | 53.8 |
| GRPO – 12 rollouts | 50.9 | 52.7 | 51.8 |
| SFT (Answer) | 58.6 | 38.7 | 48.7 |
| **MinervaRL** | 63.2 | **63.3** | **63.3** |
| **Ablations (remove component from MinervaRL)** | | | |
| EMA-off | 63.1 | 62.2 | 62.6 |
| Filtering-Off | 63.0 | 61.4 | 62.2 |
| MLFilterOff | **64.2** | 59.0 | 61.6 |

**Instruct** is most frequently preferred, suggesting that strong general instruction tuning remains a competitive baseline for response quality. Among the CTI-adapted variants, **MinervaRL** is preferred more often than **Llama-Primus-Merged**, **Foundation-Sec-8B-Instruct**, and **GRPO-only**, while the **GRPO-only** model is least preferred. This highlights the value of MinervaRL's answer-conditioned trace distillation and filtering stages for producing responses that are easier to read and verify.

### 6.4. Performance on Other Benchmarks

Table 3 shows that CTI-specific adaptation largely preserves general capability and instruction-following (MMLU-Pro (Wang et al., 2024) and IFEval (Zhou et al., 2023) remain stable across variants). At the same time, MinervaRL improves performance on CanaryExploit (a capture-the-flag-style exploit benchmark) (Bhatt et al., 2024) without a corresponding increase on WMDP-Cyber (Li et al., 2024), where lower scores indicate lower measured cyber-risk.

## 7. Ablation & Training Dynamics

**Baselines and ablations.** Table 4 compares MinervaRL against (i) the base model, (ii) GRPO, and two additional baselines: **GRPO – 12 rollouts**, which increases the per-prompt rollout budget from $N = 8$ to $N = 12$ to test whether improved exploration alone closes the gap, and **SFT (Answer)**, which directly fine-tunes the model to output the final answer on the training prompts without RL. We also

*Table 5.* Effect of self-distillation learning-rate scale $\gamma$.

| $\gamma$ | Minerva-Dev | AthenaBench-Mini | Avg |
|---|---|---|---|
| 0.01 | 60.3 | 61.3 | 60.8 |
| 0.02 | 63.1 | 63.3 | 63.2 |
| **0.05** | **63.2** | **63.3** | **63.3** |
| 0.10 | 60.1 | 58.1 | 59.1 |

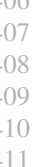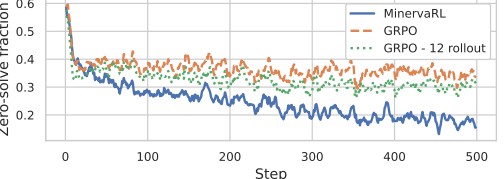

*Figure 4.* Fraction of training prompts with *no* successful rollout (max reward = 0) over training steps for Llama-8B-Instruct. Curves are smoothed with a 5-step rolling mean.

ablate key MinervaRL components by disabling the EMA teacher (EMA-off) or the filtering pipeline used to select distillation traces (Filtering-Off / MLFilterOff).

**Sensitivity to the distillation learning-rate scale.** MinervaRL performs periodic supervised distillation updates with learning rate scaled by $\gamma$ relative to the base RLVR learning rate. Table 5 shows that performance is sensitive to this knob: too small a scale under-utilizes the accepted traces, while too large a scale can destabilize training. In our setting, $\gamma = 0.05$ performs best on average, and we use this value in the main experiments.

**Reward sparsity.** A common RLVR failure mode is sampling *no* verifier-passing rollout for a given prompt (i.e., max reward = 0), resulting in an uninformative update. Figure 4 shows MinervaRL substantially reduces the fraction of prompts with zero successful rollouts compared to GRPO. Increasing GRPO's rollout budget from $N = 8$ to $N = 12$ does not achieve a comparable reduction, suggesting MinervaRL improves the policy's ability to reach valid outputs rather than merely benefiting from more sampling.

**Effect of Filtering.** Figure 5 tracks the ACR distillation pipeline over training, reporting the fraction of batches that pass the heuristic filter, pass the learned (ML) filter, and satisfy UID coverage. These fractions increase over time, indicating that as the policy improves it produces more verifier-passing rollouts and a larger share of candidates meet the quality/coverage constraints needed to form stable self-distillation traces.

**Entropy.** Figure 6 shows that MinervaRL retains relatively higher policy entropy than GRPO during training. This helps maintain exploration and reduces the risk of prematurely collapsing onto narrow response patterns, which

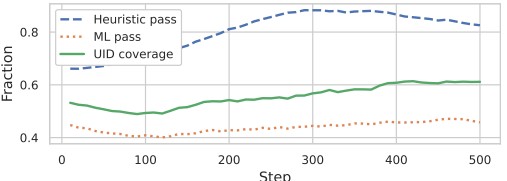

*Figure 5.* Acceptance rates of the ACR self-distillation pipeline over training for Llama-8B-Instruct: fraction of batches passing the heuristic filter, passing the learned (ML) filter, and meeting unique identifier (UID) coverage (i.e., accepted traces span a minimum number of distinct UIDs).

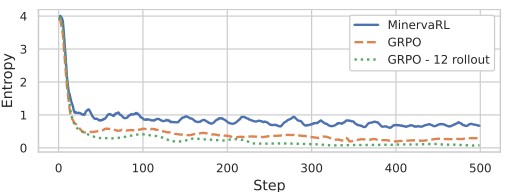

*Figure 6.* Policy entropy over training for Llama-8B-Instruct.

is particularly important under sparse verifiable rewards.

## 8. Limitations and Future Work

Our study has several limitations. **Compute and ablations:** due to compute constraints, we only study sensitivity to the distillation learning-rate scale $\gamma$, and do not sweep other key hyperparameters such as the distillation interval $I$, per-interval batch cap $M$, or EMA teacher decay $\alpha$, which may affect stability and final performance. **Data coverage:** Minerva-CTI and our evaluation suite are largely English-centric; extending to multilingual CTI and region-specific reporting styles is an important direction. **Filtering and judging:** we filter answer-conditioned reasoning traces using a lightweight TextCNN classifier; stronger LLM-judge filtering may improve quality but would increase cost, motivating scalable alternatives for trace-quality estimation. **Training cost:** MinervaRL introduces answer-conditioned rollouts and periodic distillation, increasing wall-clock time (e.g., Llama-3.1-8B-Instruct: $\sim$401$\rightarrow$550 minutes, $\sim$37%).

## 9. Conclusion

We introduced Minerva, a reinforcement learning framework with verifiable rewards for cyber threat intelligence (CTI) LLMs. Minerva combines on-policy RLVR (via GRPO) with hardness-gated answer-conditioned reasoning and periodic distillation (MinervaRL) to leverage sparse, verifier-based feedback for structured CTI outputs. Across 12 CTI benchmarks, MinervaRL consistently improves performance over strong instruction-tuned and security SFT baselines, demonstrating that verifiable-reward training is an effective and scalable approach for specializing open-weight LLMs to analyst-facing CTI tasks.

## Impact Statement

This work aims to improve open-source large language models (LLMs) for cyber threat intelligence (CTI) analysis. Our dataset and training objectives are curated for defensive CTI workflows, emphasizing structured extraction and reasoning over defensive artifacts (e.g., indicators, vulnerabilities, and mitigations) with verifier-checkable targets. Like other advances in CTI automation, the methods could be dual use: improved capability may also lower the cost of generating offensive guidance. Given the defensive orientation of our training data and tasks, we expect the resulting models to be most useful for CTI analysts working on defense.

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

## A. Minerva-CTI Dataset

Minerva-CTI comprises 16 tasks covering vulnerability descriptions, detection content, and structured threat knowledge bases. Each task is defined as an input–target prediction problem derived from a specific upstream source, with targets expressed as canonical identifiers (e.g., ATT&CK technique, tactic, or mitigation IDs). Across all tasks, the dataset contains 32,000 training instances and 1,200 validation instances. Table 6 summarizes the per-task sample counts for the training and validation splits in Minerva-CTI.

**1. Mappings-Explorer CVE→ATT&CK Exploitation.** This task maps vulnerability descriptions to the ATT&CK technique directly used for exploitation. Given a CVE description as input, the model predicts a single ATT&CK technique identifier (formatted as `Txxxx` or `Txxxx.yyy`). Ground-truth labels are obtained from the Center for Threat-Informed Defense Mappings Explorer CVE→ATT&CK mappings, with technique definitions aligned to the ATT&CK catalog (Center for Threat-Informed Defense, 2026; MITRE Corporation, 2026d).

**2. Mappings-Explorer CVE→ATT&CK Primary Impact.** This task focuses on the main adversarial impact resulting from successful exploitation. The input is the CVE description, and the target is a single ATT&CK technique identifier corresponding to the primary post-exploitation effect (e.g., credential access or privilege escalation). Labels are sourced from the Mappings Explorer and normalized using the ATT&CK technique taxonomy (Center for Threat-Informed Defense, 2026; MITRE Corporation, 2026d).

**3. Mappings-Explorer CVE→ATT&CK Secondary Impact.** This task predicts a subsequent impact enabled by the primary impact. The input consists of the CVE description concatenated with the given primary-impact technique ID, and the target is a single ATT&CK technique identifier representing the secondary effect. Ground-truth annotations are derived from the Mappings Explorer and mapped to canonical ATT&CK identifiers (Center for Threat-Informed Defense, 2026; MITRE Corporation, 2026d).

**4. Sigma→ATT&CK Tactics.** This task infers high-level adversary intent from detection logic. Given a Sigma rule excerpt—including the rule title, `logsource`, and `detection` fields—the model predicts a multi-label set of ATT&CK tactic identifiers (formatted as `TA000x`). Sigma rules are sourced from the public Sigma repository, and annotations are expressed using the ATT&CK tactic taxonomy (SigmaHQ, 2026; MITRE Corporation, 2026d).

**5. Sigma→ATT&CK Technique.** This task maps detection logic to the specific adversarial behavior it is designed to identify. Using the same Sigma rule excerpt as input, the model predicts a single ATT&CK technique identifier (formatted as `Txxxx` or `Txxxx.yyy`). Rules are drawn from the Sigma repository, with targets aligned to canonical ATT&CK technique identifiers (SigmaHQ, 2026; MITRE Corporation, 2026d).

**6. Atomic Red Team→ATT&CK Technique.** This task maps adversary procedure descriptions to their corresponding ATT&CK techniques. The input is an Atomic Red Team procedure snippet that includes execution steps or commands and platform context, and the target is a single ATT&CK technique identifier. Examples are drawn from the Atomic Red Team repository, which is natively aligned with the ATT&CK framework (Red Canary, 2026; MITRE Corporation, 2026d).

**7. Microsoft Sentinel→ATT&CK Technique.** This task links analytics rules to the ATT&CK techniques they are intended to detect. The input comprises the rule title, description, and associated KQL query, and the target is a single ATT&CK technique identifier. Rules are sourced from the Microsoft Sentinel content repository, with labels normalized to the ATT&CK taxonomy (Microsoft, 2026; MITRE Corporation, 2026d).

**8. Splunk Security Content→ATT&CK Technique.** This task maps SPL-based detection content to the ATT&CK technique it targets. The input consists of an SPL query, its detection narrative, and metadata, and the target is a single ATT&CK technique identifier. Content is obtained from Splunk Security Content, with annotations expressed using canonical ATT&CK technique IDs (Splunk, 2026; MITRE Corporation, 2026d).

**9. ATT&CK Scenario→Technique.** This task identifies the ATT&CK technique that best corresponds to a described adversary scenario. The input is a scenario text derived from ATT&CK procedure examples, and the target is a single

Table 6. Minerva training datasets and split sizes.

| Dataset | Target | Train | Val |
|---|---|---|---|
| Mappings-Explorer CVE→ATT&CK Exploitation | ATT&CK technique ID | 245 | 20 |
| Mappings-Explorer CVE→ATT&CK Primary Impact | ATT&CK technique ID | 210 | 20 |
| Mappings-Explorer CVE→ATT&CK Secondary Impact | ATT&CK technique ID | 64 | 10 |
| Sigma→ATT&CK Tactics | ATT&CK tactic IDs | 950 | 50 |
| Sigma→ATT&CK Technique | ATT&CK technique ID | 950 | 50 |
| Atomic Red Team→ATT&CK Technique | ATT&CK technique ID | 950 | 50 |
| Microsoft Sentinel→ATT&CK Technique | ATT&CK technique ID | 950 | 50 |
| Splunk Security Content→ATT&CK Technique | ATT&CK technique ID | 280 | 20 |
| ATT&CK Scenario→Technique | ATT&CK technique ID | 7,780 | 220 |
| ATT&CK Scenario→Tactics | ATT&CK tactic IDs | 1,950 | 50 |
| ATT&CK Scenario→Mitigations | ATT&CK mitigation IDs | 7,780 | 220 |
| NVD CVE→CWE | CWE IDs | 6,696 | 220 |
| NVD CVE→CVSS v3.1 | CVSS v3.1 vector | 1,900 | 100 |
| ATT&CK Threat Actor Attribution | Threat actor name | 779 | 60 |
| CAPEC Example→CAPEC | CAPEC ID | 340 | 40 |
| CAPEC Example→CWE | CWE IDs | 176 | 20 |
| Total | – | 32,000 | 1,200 |

ATT&CK technique identifier. Scenarios and labels are sourced directly from the ATT&CK knowledge base and its machine-readable releases (MITRE Corporation, 2026d;a).

**10. ATT&CK Scenario→Tactics.** This task infers the adversary intent categories implied by a scenario. Given the same scenario text as input, the model predicts a multi-label set of ATT&CK tactic identifiers (`TA000x`) associated with the underlying behavior. Annotations are derived from the ATT&CK taxonomy and its structured releases (MITRE Corporation, 2026d;a).

**11. ATT&CK Scenario→Mitigations.** This task predicts mitigations relevant to the behaviors described in a scenario. The input is the ATT&CK scenario text, and the target is a multi-label set of ATT&CK mitigation identifiers (`Mxxxx`) associated with the corresponding techniques. Mitigation mappings are obtained from the ATT&CK knowledge base (MITRE Corporation, 2026d;a).

**12. NVD CVE→CWE.** This task maps vulnerability descriptions to their underlying weakness categories. The input is the CVE description text, and the target is a multi-label set of CWE identifiers (formatted as `CWE-xxx`). CVE records are obtained from the National Vulnerability Database, with labels aligned to the CWE taxonomy (Byers et al., 2022; MITRE Corporation, 2026c).

**13. NVD CVE→CVSS v3.1.** This task predicts the CVSS v3.1 base vector associated with a vulnerability. Given a CVE description as input, the model outputs the corresponding CVSS v3.1 base vector string (e.g., `CVSS:3.1/AV:N/...`). CVE entries are sourced from the National Vulnerability Database, and targets follow the official CVSS v3.1 specification (Byers et al., 2022; FIRST.org, Inc., 2019).

**14. ATT&CK Threat Actor Attribution.** This task performs threat actor attribution from observed behaviors expressed as procedure text. Examples are derived from MITRE ATT&CK Enterprise intrusion-set (group) entries by collecting each actor's *techniques used* relationships and extracting the associated procedure descriptions that characterize how the actor operates (MITRE Corporation, 2026d;a). Training instances are sampled from per-actor pools of procedures, with counts allocated roughly in proportion to available procedure coverage (e.g., binning by technique count and enforcing minimum coverage) to prevent a small number of well-documented actors from dominating the dataset. To reduce lexical leakage, prompts are anonymized by replacing explicit actor mentions with a generic placeholder (e.g., "A threat actor") and by generalizing other named entities by type (e.g., campaign, malware, tool) while preserving the remaining structure. The true actor name is retained only as the target label, and aliases are recorded in a lookup table for evaluation and reward scoring.

**15. CAPEC Example→CAPEC.** This task maps attack example narratives to the CAPEC attack pattern they exemplify. The input is a CAPEC example description, and the target is a single CAPEC identifier (formatted as `CAPEC-xxx`). Both example texts and pattern identifiers are sourced from the CAPEC catalog (MITRE Corporation, 2026b).

**16. CAPEC Example→CWE.** This task associates CAPEC attack examples with the underlying software weakness categories they exploit. The input is the same CAPEC example narrative, and the target is a multi-label set of CWE identifiers (formatted as `CWE-xxx`). Example descriptions are drawn from CAPEC, with labels aligned to the CWE taxonomy (MITRE Corporation, 2026b;c).

## B. Reward Functions for RLVR

Each task in Minerva-CTI is paired with a task-specific reward function determined by its output structure. Given a model completion and ground-truth label $t$, we first extract a structured prediction $y$ (or a prediction set $P$) from the completion using type-specific parsing rules. While some tasks use binary rewards, others support graded credit that reflects degrees of correctness under structured output constraints.

**System Prompt and Answer Extraction.** All Minerva-CTI tasks share a common system prompt that instructs the model to reason step by step and to place its final prediction inside `\boxed{...}` to enable reliable parsing:

> Train System Prompt
>
> You are given a cyber threat intelligence question. Solve it by reasoning step by step.
> Present the final answer clearly inside \boxed{}.

For reward computation, we extract an answer span by prioritizing the final `\boxed{...}` occurrence when present. Extraction behavior differs by split. During training, parsing is strict: if no boxed span is found, we only accept explicit answer lines (e.g., `Answer:` or `Final answer:`), and we require a single unambiguous identifier for single-label tasks or a set of identifiers taken from that answer line for multi-label tasks. During validation and testing, parsing is more permissive: if no boxed span is found, we attempt answer-line parsing, then fall back to the last non-empty line or to tagged answer spans, applying regex-based matching across the response to recover the predicted identifiers.

**Normalization.** We apply light canonicalization before scoring. The function $\mathrm{norm}_{\mathrm{id}}(\cdot)$ trims surrounding whitespace and uppercases all identifiers. For ATT&CK technique IDs, we additionally standardize sub-technique notation by zero-padding the suffix to three digits when present (e.g., `T1059.3 ↦ T1059.003`). For threat actor names, we use $\mathrm{norm}_{\mathrm{actor}}(\cdot)$, which lowercases the string, strips non-alphanumeric characters, and collapses consecutive whitespace into a single space.

**Single Identifier (Exact Match).** We use an exact-match reward for tasks with a single canonical identifier output (e.g., CAPEC Example→CAPEC). Let $y$ denote the extracted prediction and $t$ the ground-truth identifier. The reward is

$$r = \mathbb{1}\left[\mathrm{norm}_{\mathrm{id}}(y) = \mathrm{norm}_{\mathrm{id}}(t)\right].$$

**ATT&CK Technique ID.** This reward is used for tasks whose target is an ATT&CK technique identifier, including CVE→ATT&CK (exploitation, primary impact, secondary impact), ATT&CK procedure scenario→technique, and detection→technique tasks (Sigma, Atomic Red Team, Microsoft Sentinel, and Splunk). Let $b(\cdot)$ extract the base technique (e.g., `T1059`) and let $s(\cdot) \in \{0, 1\}$ indicate whether an identifier includes a sub-technique suffix. We award full credit for an exact ID match after normalization. If the base technique matches but the prediction and target differ only in sub-technique specificity, we award half credit:

$$r = \begin{cases} 1.0, & \mathrm{norm}_{\mathrm{id}}(y) = \mathrm{norm}_{\mathrm{id}}(t), \\ 0.5, & b(y) = b(t) \ \wedge \ \big(s(y) = 1 \ \vee \ s(t) = 1\big), \\ 0.0, & \text{otherwise.} \end{cases}$$

This graded scoring reflects that some upstream sources annotate at the base-technique level while others use sub-technique IDs.

**Multi-label Identifier Sets.** This reward is used for tasks with set-valued targets, including ATT&CK procedure scenario→tactics, ATT&CK procedure scenario→mitigations, NVD CVE→CWE, and CAPEC Example→CWE. Let $P$ denote the normalized set of predicted identifiers (all extracted IDs) and $T$ the normalized ground-truth set. We score predictions using the set-$F_1$ measure, which penalizes both missing and spurious labels:

$$r = \begin{cases} 1.0, & P = T = \varnothing, \\ 0.0, & P = \varnothing \text{ xor } T = \varnothing, \\ \frac{2|P \cap T|}{|P|+|T|}, & \text{otherwise.} \end{cases}$$

Invalid identifiers are discarded during parsing, and repeated identifiers are removed by set construction.

**CVSS v3.1 Base Vector.** This reward is used for NVD CVE→CVSS v3.1. We parse the model output as a CVSS v3.1 vector and require that each of the eight Base metrics appear exactly once: $\mathcal{M} = \{\text{AV}, \text{AC}, \text{PR}, \text{UI}, \text{S}, \text{C}, \text{I}, \text{A}\}$. Base metrics may appear in any order. Temporal and Environmental metrics, if present, are ignored. If parsing fails due to an incorrect prefix/version, missing or duplicate Base metrics, or invalid values, the reward is set to 0.

If parsing succeeds, we compute the CVSS v3.1 Base Score $s(\cdot) \in [0, 10]$ from the parsed Base metrics and assign reward based on score distance:

$$r = \max\left(0, 1 - \frac{|s(y) - s(t)|}{\delta}\right), \qquad \delta = 10.$$

**Threat Actor Attribution.** This reward is used for ATT&CK threat actor attribution. Although each instance has a canonical actor label, we score predictions against the actor's alias set derived from ATT&CK group profiles, which includes the canonical name and known aliases. Let $A(t)$ denote this alias set for ground truth $t$, normalized with $\text{norm}_{\text{actor}}(\cdot)$. From the model's extracted answer span, we form a set of candidate names $Y$ by splitting on commas/semicolons, removing empty strings, and applying the same normalization. We assign reward if any predicted candidate matches any known alias:

$$r = \mathbb{1}[Y \cap A(t) \neq \varnothing].$$

## C. Answer-Conditioned Reasoning (ACR) Filtering

### C.1. Generation

We generate answer-conditioned reasoning (ACR) traces by augmenting each training prompt with a label-revealing block that asks the model to produce a short justification while avoiding explicit leakage language. The template is shown in Figure 7.

### C.2. Filtering

In MinervaRL, answer-conditioned reasoning (ACR) generation produces, for each training prompt (UID) $i$, a set of $K$ rollouts $\{r_{i,k}\}_{k=1}^K$. We retain at most one rollout per UID as a self-distillation (SFT) target. Each rollout is first evaluated by the task-specific Minerva verifier, yielding a base correctness score $s_{i,k} \in [0, 1]$. We keep only rollouts with $s_{i,k} = 1$ as candidates for distillation and then apply a two-stage filtering pipeline to select a single high-quality target among the remaining rollouts.

**Heuristic filtering.** We filter out rollouts that exhibit leakage or low-quality generation artifacts using four checks: (i) *explicit leakage cues*, detected by a curated list of phrases and regex patterns indicating direct references to labels or provided answers; (ii) *insufficient reasoning*, where the reasoning portion (all lines except the final answer line) has fewer than 100 characters; (iii) *low semantic grounding*, where the Jaccard overlap is below 0.05 between reasoning tokens and the concatenation of the task description plus label reference (computed over lowercased alphanumeric tokens, with ID-like tokens removed); and (iv) *degenerate responses*, detected using repetition heuristics applied only when the response length is at least 30 tokens. Degeneracy checks include $n$-gram repetition thresholds (reject if $\text{rep}_3 \geq 0.70$ or $\text{rep}_4 \geq 0.75$), repeated-window checks (window size 24, stride 12; reject on exact window repeats or if 3-gram Jaccard similarity is $\geq 0.9$ across non-overlapping windows), and near-duplicate sentence checks (within a 6-sentence window, considering sentences with at least 6 words; reject if SequenceMatcher similarity is $\geq 0.75$). Candidates that pass all heuristic checks are forwarded to an ML-based quality filter.

---

**ACR Generation Prompt**

You are generating a reasoning trace for training.

GROUND_TRUTH_LABELS:
- <LABEL_1>
- <LABEL_2>

LABEL_REFERENCE:
<details_text>

Instructions:
- Write a short reasoning that would justify selecting the correct label(s) from the input.
- <TASK_OR_ENTITY_REASONING_HINT>
- Do NOT say or imply that the answer was provided (no phrases like "given the answer", "based on the provided label", "ground truth", etc.).
- End with the final answer in the same format required by the original task.

---

*Figure 7.* Answer-conditioned reasoning (ACR) prompt template used to elicit training traces.

**ML quality filter and selection.** We score the remaining candidates with a lightweight TextCNN classifier that operates on response-only alphanumeric tokens, truncates to at most 1024 tokens, and outputs a "good" probability $m_{i,k} \in [0, 1]$. We keep candidates with $m_{i,k} \geq \tau_m$ (we use $\tau_m = 0.5$), yielding the final eligible set

$$\mathcal{E}_i = \{ k \mid s_{i,k} = 1, \ L_{i,k} = 0, \ D_{i,k} = 0, \ m_{i,k} \geq \tau_m \},$$

where $L_{i,k}$ indicates heuristic leakage rejection and $D_{i,k}$ indicates degeneracy rejection. If $\mathcal{E}_i = \varnothing$, we do not distill an example for UID $i$. Otherwise, we select

$$k^\star = \max\Big\{ k \in \mathcal{E}_i \ : \ m_{i,k} = \max_{j \in \mathcal{E}_i} m_{i,j} \Big\}.$$

breaking ties uniformly at random. The resulting distillation record pairs the original prompt (without ACR hints) with the selected rollout response.

## D. Training the TextCNN Reasoning-Quality Classifier

MinervaRL uses a lightweight TextCNN classifier as an optional ML filter for ACRD self-distillation, aimed at removing low-quality reasoning traces while preserving answer correctness. We train the classifier on a GOOD/BAD dataset where labels reflect reasoning quality (e.g., leakage, incoherence, ungrounded claims) rather than correctness.

### D.1. Dataset Curation

We start from the Minerva-CTI training prompts and generate multiple responses per prompt in two variants: (i) *plain*, using the original prompt only, and (ii) *hinted*, where we augment the prompt with an ACR block that provides (a) ground-truth labels, (b) a label reference, and (c) an explicit marker indicating that the model is producing a reasoning trace for training. Responses are generated using a diverse set of open and open-weight instruction-tuned LLMs, with typical decoding settings of temperature $0.7$ and a maximum length of 1024 new tokens.

To isolate reasoning quality from answer correctness, each response is scored using the task verifier and we keep only *reward-correct* responses, defined as `reward = 1`. These correct responses are then labeled GOOD/BAD by a rubric-based LLM judge.

We label reward-correct responses with a rubric-based LLM judge that assigns each response a binary label (`GOOD` or `BAD`) based on reasoning quality and leakage, without re-grading answer correctness. The judge is instructed to return exactly one JSON object on a single line. The prompt template used for judge labeling is shown in Figure 8.

---

**LLM Judge Prompt**

You are an expert CTI evaluator. You will be given a Question and a Response.
The Question may include answer/reference text as hints.

Label the Response as GOOD or BAD. Mark BAD if any single criterion below is present;
otherwise mark GOOD. Do not grade correctness beyond these checks.

Return exactly one JSON object on a single line and nothing else.
- GOOD: {"label": "GOOD"}
- BAD: {"label": "BAD", "category_id": <number>, "category_title": "<title>"}

If multiple BAD criteria apply, choose the lowest-numbered category.

Criteria (one per bullet):
1. Leakage: says or implies the answer/label/options/reference were provided,
   or quotes/paraphrases provided reference text instead of reasoning from the prompt.
2. Incoherent: loops, repeated phrases/lines, templated filler, or gibberish.
3. Ungrounded: invents concrete details not in the question (extra CVEs,
   vendors, malware names, IOCs, dates, techniques, etc.).
4. Mismatch: reasoning supports a different label than the final answer,
   or directly contradicts it.
5. Unsupported: provides reasoning but does not use evidence from the prompt to
   justify the answer (hand-wavy or irrelevant justification), OR provides an
   answer with no reasoning at all (answer-only).
6. Other: refusals, policy/meta artifacts, generic CTI tutorials, or prompt copying.

Example outputs:
{"label": "GOOD"}
{"label": "BAD", "category_id": 1, "category_title": "Leakage"}

QUESTION:
{QUESTION}

RESPONSE:
{RESPONSE}

---

*Figure 8.* Prompt template used to label reward-correct responses as GOOD/BAD for training the TextCNN reasoning-quality classifier.

The judge marks a response as BAD if any single failure criterion is present, including: (1) *Leakage* (explicitly stating or implying that the answer/label/reference was provided, or copying/paraphrasing the reference), (2) *Incoherent* (loops, repeated filler, or gibberish), (3) *Ungrounded* (inventing concrete details not supported by the question), (4) *Mismatch* (reasoning contradicts the final answer), (5) *Unsupported* (hand-wavy justification or answer-only), and (6) *Other* (refusals, meta artifacts, generic tutorials, or prompt copying). To increase coverage of BAD modes, we additionally synthesize BAD responses by conditioning a generator on a randomly sampled rubric violation; synthetic examples are retained only if they remain reward-correct under the verifier. Finally, we construct a balanced dataset by downsampling to 32k GOOD and 32k BAD examples and split each class 80/20 into train/validation partitions.

### D.2. TextCNN Model and Training

We train a standard TextCNN classifier over tokenized text. Tokens are extracted with a regex tokenizer `[A-Za-z0-9_]+` and lowercased. We consider two input formats: *response-only* (used in ACRD filtering) and *response+prompt*, which concatenates response tokens, a separator token `sep`, and prompt tokens; when a maximum token budget is enforced, response tokens are preserved when possible and prompt tokens are truncated to fit. The vocabulary is built from the training split with `max_vocab`=100,000 and `min_freq`=2, using `<pad>` and `<unk>` as special tokens.

The architecture consists of an embedding layer, followed by 1D convolutions with multiple kernel widths (default 3/4/5), ReLU activations, max-over-time pooling, feature concatenation, dropout, and a 2-way linear classifier. We train with

*Table 7.* Evaluation datasets used for Table 1 (meta tasks report the sum over subtasks).

| Task | # Samples | Output format (summary) |
| --- | --- | --- |
| CKT (Alam et al., 2025) | 3000 | Multiple-choice CTI questions; output a single letter A–E. |
| CyberMetric (Tihanyi et al., 2024) | 2000 | Multiple-choice cybersecurity questions; output a single letter A–D. |
| SOCEval (Deason et al., 2025) | 588 | Multi-select SOC-style reasoning over threat intel reports; output a JSON object in `<json_object>` tags with a `correct_answers` array. |
| RCM (Alam et al., 2025) | 2000 | Root cause mapping; output a single CWE identifier `CWE-####` |
| VSP (Alam et al., 2025) | 2000 | CVSS v3.1 base vector prediction; output a full vector string (e.g., `CVSS:3.1/AV:N/...`). |
| ATE (Alam et al., 2025) | 500 | Attack technique extraction; output a single technique ID `T####` |
| RMS (Alam et al., 2025) | 500 | Risk mitigation strategy; output a set of mitigation IDs `M10xx`. |
| ElasticRule | 432 | Map an Elastic detection rule to a single ATT&CK technique ID `T####`. |
| APTNER (Wang et al., 2022) | 1505 | APT-focused NER; output a JSON object mapping entity types to lists of strings. |
| LANCE (Prism meta) (Froudakis et al., 2025) | 466 | IoC identification over candidate indicators (IP/URL/Domain/Hash); output labels per candidate as specified by the prompt. |
| AnnoCTR (meta) (Lange et al., 2024) | 1230 | STIX-style entity/relation extraction; outputs use task-specific XML-like tags (e.g., `<entities>`, `<related>`, `<label>`). |
| AZERG (meta) (Lekssays et al., 2025) | 1333 | STIX-style entity/relation extraction; outputs use task-specific XML-like tags (e.g., `<entities>`, `<related>`, `<label>`). |

AdamW and cross-entropy loss for 5 epochs with batch size 128, reporting validation accuracy/precision/recall/$F_1$ after each epoch. We select hyperparameters via a grid search over learning rate $\{3\times10^{-4}, 6\times10^{-4}, 10^{-3}, 2\times10^{-3}, 4\times10^{-3}\}$, filters per kernel $\{256, 384\}$, dropout $\{0, 0.25\}$, embedding dimension $\{200, 300\}$, and max token budgets $\{2048, 3072\}$. The best model in our sweep uses learning rate $6\times10^{-4}$, kernels 3/4/5, 384 filters per kernel, embedding dimension 200, max tokens 3072, and dropout 0.25, achieving a validation accuracy of 0.828.

# E. Evaluation Datasets

We evaluate models on a suite of CTI and cybersecurity benchmarks spanning multiple-choice QA, structured taxonomy prediction (ATT&CK/CWE/CVSS), and information extraction. Each task is evaluated with task-specific answer extraction and exact/structured matching.

**Task descriptions (Table 1 order).**

1. **CKT** (Alam et al., 2025): Cyber Threat Intelligence multiple-choice QA. Input is a prompt with five options (A–E); output is a single letter A–E on the final line (optional brief justification allowed).

2. **CyberMetric** (Tihanyi et al., 2024): general cybersecurity multiple-choice QA. Input is a prompt with four options (A–D); output is a single letter A–D on the final line.

3. **SOCEval** (Deason et al., 2025): multi-select reasoning over threat intel reports. Input includes report context and a question with options; output is a JSON object wrapped in `<json_object>` tags containing a `correct_answers` list.

4. **RCM** (Alam et al., 2025): root-cause mapping from CVE to CWE. Input is a CVE description; output is a single `CWE-####` identifier on the final line.

5. **VSP** (Alam et al., 2025): vulnerability scoring prediction. Input is a CVE description; output is a full CVSS v3.1 vector string (e.g., `CVSS:3.1/AV:N/...`).

6. **ATE** (Alam et al., 2025): ATT&CK technique extraction. Input describes attacker behavior (Windows environment); output is a single technique ID `T####` or `T####.###` on the final line.

7. **RMS** (Alam et al., 2025): risk mitigation strategy. Input describes an attack scenario; output is a set of mitigation IDs `M10xx`.

8. **ElasticRule:** Elastic rule to ATT&CK technique mapping. Input contains a detection rule (query + metadata); output is a single ATT&CK technique ID `T####`.

9. **APTNER** (Wang et al., 2022): APT-focused named-entity recognition. Input is a sentence with entity type definitions; output is a JSON object mapping entity types to extracted entities.

10. **LANCE** (Froudakis et al., 2025): IoC identification (Prism meta task). Input is a report segment with a list of candidate indicators; output labels each candidate as IoC vs. non-IoC (aggregated over IP/URL/Domain/Hash subtasks).

11. **AnnoCTR** (Lange et al., 2024): STIX-style entity and relation extraction. This meta task aggregates four subtasks (entity extraction, entity typing, relation existence, relation label) with XML-like tag formats.

12. **AZERG** (Lekssays et al., 2025): STIX-style entity and relation extraction. This meta task aggregates four subtasks (entity extraction, entity typing, relation existence, relation label) with XML-like tag formats.

# F. Additional Theory: Why MinervaRL Can Expand Empirical Support

**Setup (answer-level view).** Each training instance is $(x, a^\star)$, where $x \in \mathcal{X}$ is the original prompt and $a^\star \in \mathcal{A}$ is the ground-truth structured answer (e.g., an ATT&CK technique ID, a tactic set, a CVSS vector, etc.). Let $g : \mathcal{Y} \to \mathcal{A}$ be the task extractor mapping a full completion $y \in \mathcal{Y}$ to an extracted answer $a = g(y)$.

To mirror the main text, we write the (binary) verifiable reward as

$$R_{\mathrm{minerva}}(x, y; a^\star) := \mathbb{1}\!\left[g(y) = a^\star\right],$$

noting that task-specific partial credit can be accommodated by thresholding at correctness. A policy $\pi_\theta(\cdot \mid x)$ over completions induces an *answer distribution*

$$p_\theta(a \mid x) := \Pr_{y \sim \pi_\theta(\cdot \mid x)}\left[g(y) = a\right].$$

In particular, $p_\theta(a^\star \mid x) = \Pr_{y \sim \pi_\theta}[R_{\mathrm{minerva}}(x, y; a^\star) = 1]$.

**Empirical support under a rollout budget.** Following the empirical-support viewpoint, define a detectability threshold for $k$ i.i.d. samples and failure probability $\zeta \in (0, 1)$:

$$\varepsilon_{k,\zeta} := \frac{-\log \zeta}{k}.$$

Intuitively, answers with probability below $\varepsilon_{k,\zeta}$ are unlikely to be observed in $k$ rollouts.

**Lemma F.1** (Finite-sample detectability bound). *Fix a prompt $x$ and an answer $a \in \mathcal{A}$. Let $p = p_\theta(a \mid x)$. If $p \geq \varepsilon_{k,\zeta}$, then the probability that $k$ independent rollouts miss $a$ is at most $\zeta$:*

$$\Pr[a \text{ is not observed in } k \text{ rollouts}] = (1 - p)^k \leq \zeta.$$

*Equivalently, if $a$ is never observed in $k$ rollouts, then with confidence at least $1 - \zeta$ we have $p \leq \varepsilon_{k,\zeta}$.*

**Why small-budget RLVR can stall on "zero-reward" prompts.** Consider a training regime that (i) uses at most $k$ rollouts per prompt per iteration, and (ii) performs on-policy RL updates using only those sampled trajectories and their rewards. If $p_\theta(a^\star \mid x) \ll 1/k$, then successes on $x$ are rare; the waiting time to see the first success scales like $1/\mathrm{pass@}k$.

**Theorem F.2** (Small-budget sampling barrier for on-policy RLVR (per-prompt)). *Fix a prompt $x$ and rollout budget $k$. Let $p_t := p_{\theta_t}(a^\star \mid x)$ be the policy's answer-level success probability at iteration $t$, and suppose $p_t \leq \varepsilon_{k,\zeta}$. Then, with probability at least $1 - \zeta$, the $k$ on-policy rollouts at iteration $t$ contain* no *verified-correct trajectory for $x$. In that event, any per-prompt update rule whose only positive learning signal for $a^\star$ comes from* observed *verified-correct rollouts cannot reliably* increase *$p_{t+1}$ above $\varepsilon_{k,\zeta}$ in that iteration.*

*Proof sketch.* The first statement is Lemma F.1. For the second, condition on the (typical) event that all sampled rewards are zero for prompt $x$. Then the update has no direct per-prompt evidence that distinguishes $a^\star$ from other unobserved answers under $x$ within that iteration; hence there is no reliable mechanism to move $p_\theta(a^\star \mid x)$ above the detectability threshold without either (a) observing a success for $x$, or (b) injecting an external supervised signal tied to $(x, a^\star)$. □

**MinervaRL (answer-conditioned exposure + distillation).** MinervaRL augments RLVR with *answer-conditioned trace generation* for hard samples and *distillation onto the original prompt*. Concretely, when $k$ base rollouts on $x$ fail, MinervaRL constructs a modified prompt $\tilde{x} = \mathrm{ACR}(x, a^\star, \mathrm{ref}(a^\star))$ that reveals the gold answer $a^\star$ and (optionally) a truncated label reference. The actor samples $k_{\mathrm{acr}}$ completions on $\tilde{x}$, filters them using leakage/quality guards, and then performs SFT updates on the *original* prompt $x$ using an accepted trace.

We encode this as two assumptions: (i) answer-conditioned *exposure* produces a verified trace with nontrivial probability, and (ii) an SFT step produces a nontrivial increase in the answer-level probability on the original prompt.

**Assumption F.3** (Answer-conditioned exposure for hard prompts). For any hard instance $(x, a^\star)$ (where base rollouts on $x$ fail), there exists $\alpha > 0$ such that a single ACR attempt (sampling $k_{\mathrm{acr}}$ times from $\pi_\theta(\cdot \mid \tilde{x})$ and applying the MinervaRL acceptance filters) yields at least one accepted completion $y_{\mathrm{acr}}$ with $R(x, y_{\mathrm{acr}}) = 1$ with probability at least $\alpha$.

**Assumption F.4** (Effective distillation step (answer-level)). There exists $\Delta > 0$ such that, whenever MinervaRL performs an SFT update on an accepted pair $(x, y_{\mathrm{acr}})$ with $R(x, y_{\mathrm{acr}}) = 1$, the induced success probability increases by at least a multiplicative factor in log-space:

$$\log p_{\theta+}(a^\star \mid x) \geq \log p_\theta(a^\star \mid x) + \Delta,$$

as long as $p_\theta(a^\star \mid x)$ remains below a fixed constant (i.e., before saturation).

**Theorem F.5** (Empirical support seeding via answer-conditioned trace distillation). *Fix $(x, a^\star)$ and a small rollout budget $k$ for base RLVR. Let $\varepsilon_{k,\zeta}$ be as above. Under Assumptions F.3–F.4, MinervaRL raises the success probability $p_\theta(a^\star \mid x)$ above the detectability threshold in a finite expected number of ACR+SFT cycles. In particular:*

*1. The number of ACR attempts until the first accepted verified trace is geometric with mean $\leq 1/\alpha$.*

*2. After $N = \left\lceil \frac{\log \varepsilon_{k,\zeta} - \log p_0}{\Delta} \right\rceil$ successful distillation updates on $(x, y_{\mathrm{acr}})$, we have $p_\theta(a^\star \mid x) \geq \varepsilon_{k,\zeta}$.*

*Consequently, once $p_\theta(a^\star \mid x) \geq \varepsilon_{k,\zeta}$, the probability that the base RLVR rollouts on $x$ remain all-zero at budget $k$ is at most $\zeta$ (Lemma F.1).*

*Proof sketch.* Item (1) is immediate from Assumption F.3. Item (2) follows by iterating Assumption F.4. The final statement is Lemma F.1 applied to $a^\star$. □

**Corollary F.6** (Expected cycles to cross the detectability threshold). *With the notation of Theorem F.5, let $N = \left\lceil \frac{\log \varepsilon_{k,\zeta} - \log p_0}{\Delta} \right\rceil$. The expected number of ACR attempts needed to obtain $N$ successful distillation updates is at most $N/\alpha$ (negative-binomial), so MinervaRL crosses the detectability threshold in $\mathbb{E}[ACR\ attempts] \leq N/\alpha$ under Assumptions F.3–F.4.*

**Scope and limitations.** The result above is intentionally narrow: it explains how MinervaRL can reduce the incidence of *zero-reward* prompts under a *fixed, small rollout budget* by seeding verified trajectories and distilling them onto the original prompt. It does *not* claim that answer-conditioned traces are faithful proofs, nor that the mechanism will transfer to domains like mathematics where conditioning on the final answer may not teach the model to derive it.

## G. Training Hyperparameters and Implementation Details

This appendix lists the training hyperparameters used for RLVR (GRPO) and for MinervaRL. We report only settings that directly affect optimization, sampling, or sequence truncation.

### G.1. RLVR (GRPO) settings

- **Optimizer:** GRPO with actor learning rate $1 \times 10^{-6}$.

- **Batching:** 128 prompts per training step.

- **Rollouts:** $N = 8$ sampled completions per prompt per step.

- **Sequence lengths:** max prompt length 2048 tokens; max response length 1024 tokens.

- **Schedule:** 500 training steps.

### G.2. MinervaRL (ACR + distillation) settings

- **Hard-example criterion:** mark a prompt "hard" if the best base-rollout reward is $< 1.0$ (CVSS prompts excluded).

- **ACR prompt context:** max ACR prompt length 4096 tokens; max response length 1024 tokens.

- **ACR sampling:** $K = 4$ traces per ACR prompt; temperature 0.7; nucleus sampling $p = 0.9$.

- **Deferred generation cadence:** generate/distill every $I = 10$ steps.

- **Teacher:** EMA teacher with decay $\alpha = 0.995$.

- **Distillation:** supervised fine-tuning on original prompts using up to 256 accepted traces per distillation interval; learning rate is scaled by $\gamma = 0.05$ relative to the RLVR learning rate.

- **Trace filtering:** a two-stage pipeline (heuristics + ML filter) is applied before distillation; the ML filter acceptance threshold is $\tau_q = 0.5$.

