# OpenReview forum: "Minerva: Reinforcement Learning with Verifiable Rewards for Cyber Threat Intelligence LLMs"
_ICML.cc/2026/Conference — Submitted to ICML 2026_

### Official Review · Reviewer_sPdc · 2026-03-01

**Soundness:** 3
**Presentation:** 2
**Significance:** 2
**Originality:** 2
**Overall Recommendation:** 3
**Confidence:** 3

**Summary:**

The main contributions of this paper are including two points. First, it proposes a 16-task training suite called Minerva-CTI, which covers multiple CTI subtasks such as vulnerability mapping and detection rule mapping. Second, the authors use RLVR instead of existing methods that primarily rely on SFT. This paper describes a lightweight self-training mechanism on a GRPO-based RLVR to alleviate sparse rewards and zero-success rollouts, via hardness-gated answer-conditioned reasoning and periodic distillation back to the answer-free prompt. The proposed MinervaRL improves performance on all 12 benchmarks.

**Compliance With Llm Reviewing Policy:**

Affirmed.

**Final Justification:**

The contributions of this paper are quite thin. The responses did not fully addressed my concerns.

**Key Questions For Authors:**

1. Does the gain of MinervaRL come from RLVR, or from the additional supervision of "labeled ACR self-training and distillation"?
2. What are the training data and error patterns of the TextCNN filter?

**Limitations:**

yes

**Strengths And Weaknesses:**

**Strength**
1. The paper addresses the issue of numerous zero-reward prompts due to long-tailed IDs and limited rollout budgets. MinervaRL uses a gold-labeled ACR prompt to increase the probability of successful trajectories, then distills it back to the original prompt, improving the learnability of subsequent on-policy.
2. The authors compare GRPO-12 rollout with GRPO to demonstrate whether further exploration can match the performance of MinervaRL. Furthermore, image analysis shows that MinervaRL's improvement is not solely due to an increased number of samples.
3. The authors introduce hardness-gated ACR and distill back in MinervaRL as a clear and low-cost way to alleviate sparse rewards, without relying on a stronger teacher or incurring human costs.

**Weakness**
1. The paper uses a two-stage filter—heuristics + TextCNN ML filter—to filter ACRs. The trajectory is set, and a threshold is set simultaneously. This introduces a new bottleneck where the quality of the set filter determines the distillation quality. Furthermore, the main body currently does not provide the training data, error analysis, or causal explanation of the filter's impact on the final result. Existing explanations and descriptions are insufficient.
2. The paper may lack innovation; it feels like a combined application of ACR and self-distillation.
3. The method described in the paper explicitly exposes the gold label in the ACR prompt during training and then distills it back to the answer-free prompt. Although the authors emphasize that the label is not exposed during inference, a more rigorous distinction still needs to be made between the contributions of "optimization brought by RLVR" and "additional supervision signals/data augmentation" to the performance. The current experimental interpretation is insufficient.

---

> ### Author Rebuttal · Authors · 2026-03-31
>
> We thank the reviewers for the constructive feedback and address the main points below.
>
> **1. Performance gain.** The gain of MinervaRL comes from the combination of **RLVR + periodic self-distillation**, not from either part alone. RLVR provides the main verifier-driven learning signal and is important for generalization [1], but under limited-rollout CTI training it can frequently encounter hard prompts with no successful rollout, making the update uninformative. This is exactly the bottleneck MinervaRL is designed to address.
>
> Our self-distillation phase complements RLVR by targeting those hard cases. Specifically, MinervaRL uses answer-conditioned generation to seed verified traces for prompts that fail under standard RLVR, and then distills those traces back onto the original answer-free prompt. We support this mechanism both formally in Section 4.3 and empirically in Figure 4, where MinervaRL reduces the zero-solve fraction beyond what is achieved by simply increasing GRPO rollouts; additional plots for the other backbones are [here](https://tinyurl.com/4jfwj5t7).
>
> The ablations are consistent with this interpretation. In Table 4, full MinervaRL outperforms GRPO, while removing key MinervaRL components reduces performance. In addition, we have now added two rejection fine-tuning (RFT) baselines that use self-training/distillation without the full MinervaRL pipeline. MinervaRL consistently outperforms both GRPO and these RFT baselines, showing that the improvement is not explained by added supervision alone, but by the synergy between verifier-driven RLVR and targeted self-distillation on hard prompts; see [here](https://tinyurl.com/yc57vt8k).
>
> Thus, RLVR provides the core training signal, while labeled ACR self-training improves learning precisely in the sparse-reward regime where standard RLVR is weakest [2].
>
> **2. Filtering.** We provide the training-data construction and model details for the lightweight TextCNN filter in **Appendix D**. Appendix D describes (i) how the GOOD/BAD training set is built from verifier-correct responses, (ii) how weak labels are assigned using an LLM judge with a rubric covering leakage, incoherence, ungrounded claims, mismatch, unsupported reasoning, and related artifacts, and (iii) the TextCNN architecture and training setup. The classifier achieves 82.8% F1-score in our sweep.
>
> The validation confusion matrix is:
>
> |              | Pred: GOOD | Pred: BAD |
> |--------------|-----------:|----------:|
> | **Actual: GOOD** | 5,375 | 1,025 |
> | **Actual: BAD**  | 1,206 | 5,194 |
>
> We also provide evidence for the effect of filtering in **Table 4** through ablations. Removing the filter hurts generalization on AthenaBench-Mini relative to full MinervaRL, supporting the claim that filtering improves the quality of the traces used for distillation.
>
> **3. Innovation.** We agree that MinervaRL combines on-policy RLVR with periodic answer-conditioned self-distillation. Our novelty claim is not that either ingredient is new by itself, but that their combination directly addresses a CTI-specific bottleneck.
>
> Under limited-rollout RLVR, many hard CTI prompts produce no successful rollout because the target spaces are long-tailed and verifier constraints are strict. MinervaRL addresses this by using answer-conditioned generation only for such hard prompts, filtering verified high-quality traces, and distilling them back onto the original answer-free prompt. We support this both theoretically in Section 4.3 and Appendix F, and empirically in Figure 4. We also note that this is, to the best of our knowledge, the first CTI-focused RLVR framework built around verifier-checkable CTI tasks, rather than a direct transfer of existing RLVR recipes from math/code. The added RFT baselines further clarify this point: MinervaRL outperforms both GRPO and self-training-only variants, suggesting that the gains are not explained by RLVR alone or by additional supervision alone, but by their combination.
>
> [1] Chu, Tianzhe, et al. "SFT memorizes, RL generalizes: A comparative study of foundation model post-training." ICML 2025.
> [2] Yue, Yang, et al. "Does reinforcement learning really incentivize reasoning capacity in LLMs beyond the base model?." NeurIPS 2025.

---

> > ### Author Rebuttal · Reviewer_sPdc · 2026-04-01
> >
> > The responses did not fully address the concerns.
> >
> > 1. The response did not clarify how much each component contributes quantitatively.
> > 2. The confusion matrix shows counts but no qualitative analysis of what kinds of errors occur.
> > 3. The "first CTI-focused RLVR framework" is still a narrow novelty claim.
> >
> > I decide to keep the scores.

---

> > > ### Author Response · Authors · 2026-04-07
> > >
> > > We thank the reviewer for the follow-up. Below we address the remaining concerns.
> > >
> > > **1. Component contributions.** To isolate the role of each ingredient, we report the two main components of MinervaRL applied in isolation across the same four backbones: **GRPO** as the on-policy RLVR baseline, and **STaR-CTI** as the iterative self-training / distillation baseline. MinervaRL is best by average score on all four backbones. Relative to **GRPO**, it improves the 12-task average by **+4.2, +6.2, +6.4, and +0.4** points for Llama-3.1-8B, Llama-3.2-3B, Qwen3-8B, and Qwen3-4B, respectively. Relative to **STaR-CTI**, the gains are **+10.5, +11.0, +16.6, and +2.6**. This supports the claim that the gain does not come from RLVR alone or from distillation alone, but from their combination.
> > >
> > > | Backbone | STaR Avg | GRPO Avg | MinervaRL Avg | Δ vs GRPO | Δ vs STaR |
> > > |---|---:|---:|---:|---:|---:|
> > > | Llama-3.1-8B | 49.7 | 56.0 | 60.2 | +4.2 | +10.5 |
> > > | Llama-3.2-3B | 37.3 | 42.1 | 48.3 | +6.2 | +11.0 |
> > > | Qwen3-8B | 36.8 | 47.0 | 53.4 | +6.4 | +16.6 |
> > > | Qwen3-4B | 45.4 | 47.6 | 48.0 | +0.4 | +2.6 |
> > >
> > > In addition, **Table 4** in the paper also shows that removing EMA or filtering lowers validation performance, indicating that both components contribute to the final result.
> > >
> > > **2. Filter errors.** The previous response reported aggregate counts only; here we add the qualitative error patterns. The TextCNN filter is trained on verifier-correct responses labeled with the LLM-judge rubric described in Appendix D. Among the 1,206 false positives, the dominant error type is Leakage (738, 61.2%), followed by Ungrounded (214, 17.7%) and Mismatch (159, 13.2%). Among the 1,025 false negatives, errors are concentrated in the more open-ended ATT&CK generation tasks, especially `scenario_to_attack_mitigations` (392, 38.2%) and `scenario_to_attack_technique` (194, 18.9%).
> > >
> > > Qualitatively, the false positives are usually not incoherent traces. They are fluent responses that appear plausible but still reveal the target too directly or introduce unsupported reasoning. One example is below; the response is classifier-predicted as GOOD, but the phrase *“Considering the provided techniques”* is a subtle leakage cue:
> > >
> > > > A malware has installed a bootkit on the system to maintain persistence. A bootkit modifies the boot sectors of a hard drive, allowing malicious code to execute before the operating system has loaded. This behavior aligns with the definition of techniques that involve modifying the operating system or firmware to achieve persistence.
> > > >
> > > > The Master Boot Record (MBR) and Volume Boot Record (VBR) are key components involved in the boot process, and modifying these areas would allow an adversary to execute malicious code before the operating system loads. This is a common method used by bootkits to persist on systems.
> > > >
> > > > `Considering the provided techniques`, the most suitable ID is T1542.003, which directly corresponds to Bootkit, as it describes the use of a bootkit to persist on systems.
> > > >
> > > > Therefore, the selected technique ID is:
> > > >
> > > > \boxed{T1542.003}
> > >
> > > **3. Novelty and scope.** Our contribution has two parts. First, we build a unified CTI training dataset and establish a GRPO-based RLVR setup across multiple CTI task families (Sections 3, 4.1, Appendices A/B). Second, we introduce a training recipe that targets the CTI-specific sparse-reward bottleneck by combining on-policy RLVR with answer-conditioned distillation only for hard prompts.
> > >
> > > While MinervaRL is motivated by CTI, the algorithm itself is not inherently CTI-specific. It can also apply in other structured-target RLVR settings with similar reward sparsity from long-tail answers. To probe this, we ran a preliminary transfer study on text-to-SQL using the same training/validation data as SQL-R1 [1]. We trained GRPO and MinervaRL from `Qwen2.5-Coder-3B` under the same setup and compared them against both the base model and the released `SQL-R1-3B` checkpoint. MinervaRL achieves the best average score and the best result on four out of seven datasets:
> > >
> > > | Model | Spider Dev | Spider Test | BIRD Dev | Spider-DK | Spider-Syn | Spider-Realistic | LiveSQLBench | Avg |
> > > | --- | ---: | ---: | ---: | ---: | ---: | ---: | ---: | ---: |
> > > | Qwen2.5-Coder-3B | 77.4 | 77.6 | 50.3 | 65.6 | 67.2 | 69.7 | 3.7 | 58.8 |
> > > | SQL-R1-3B | 77.1 | 77.4 | 53.3 | 70.5 | 67.5 | 66.5 | 6.3 | 59.8 |
> > > | GRPO | 77.5 | 78.5 | 54.2 | 69.0 | 67.9 | 68.7 | 6.3 | 60.3 |
> > > | MinervaRL | 78.2 | 79.3 | 52.5 | 69.2 | 70.7 | 69.3 | 6.7 | 60.8 |
> > >
> > > This single transfer result is not meant to establish arbitrary domain generality. Rather, it suggests that the method may also be useful beyond CTI in RLVR settings where standard GRPO can stall because of sparse reward or long-tail targets.
> > >
> > > **Ref**
> > >
> > > [1] Ma, Peixian, et al. "SQL-R1: Training Natural Language to SQL Reasoning Model By Reinforcement Learning." NeurIPS 2025.

---

### Official Review · Reviewer_QS8q · 2026-03-01

**Soundness:** 3
**Presentation:** 3
**Significance:** 3
**Originality:** 3
**Overall Recommendation:** 4
**Confidence:** 3

**Summary:**

This work intends to investigate a general area of applying reinforcement learning with verifiable rewards (RLVR) to specialized cyber threat intelligence (CTI) tasks for large language models (LLMs), addressing critical limitations of existing supervised fine-tuning (SFT) approaches in generating robust, structured, and standards-aligned CTI outputs. The paper identifies that CTI tasks have inherent deterministic verifiability via canonical identifiers and schemas, but standard on-policy RLVR suffers from severe reward sparsity in this domain, as long-tail identifiers and strict schema requirements lead to frequent zero-reward rollouts. To address this, the authors introduce Minerva, a unified framework consisting of two core components: 1) Minerva-CTI, a curated dataset with 16 CTI tasks spanning vulnerability mapping, detection mapping, and threat attribution, paired with task-specific programmatic verifiers for deterministic reward scoring; 2) MinervaRL, an RLVR extension that mitigates reward sparsity via hardness-gated answer-conditioned reasoning (ACR) and periodic self-distillation, which seeds verified trajectories for hard samples and distills them back into the model without label leakage at inference. Extensive experiments across 4 open-weight LLM backbones and 12 mainstream CTI benchmarks show that MinervaRL consistently outperforms strong instruction-tuned and security-specific SFT baselines, with additional qualitative results demonstrating improved response quality for CTI analyst workflows. Overall, a central area presented by this paper is the mitigation of reward sparsity in RLVR for domain-specific structured generation tasks, paired with a comprehensive, verifiable training and evaluation suite for CTI-focused LLMs.

**Compliance With Llm Reviewing Policy:**

Affirmed.

**Final Justification:**

I appreciate the authors' detailed response, but it has not fully addressed all of my concerns; therefore, I will maintain my current score.

**Key Questions For Authors:**

1. You note in Section 8 that compute constraints prevented a sweep of key MinervaRL hyperparameters including distillation interval I, distillation batch cap M, and EMA decay α. Could you provide at least a limited sensitivity analysis for these hyperparameters, or share the reasoning behind your chosen default values and their expected robustness range?
2. Your paper only compares MinervaRL to a GRPO baseline with increased rollouts, but does not benchmark against other established methods for mitigating RLVR reward sparsity (e.g., adaptive exploration, curriculum learning, off-policy RLVR). Could you either provide additional benchmark results against these methods, or elaborate in the paper on the core advantages of MinervaRL over these approaches for CTI tasks?
3. Your qualitative preference analysis is limited to the Llama-3.1-8B backbone and uses only an LLM judge without human validation. Could you either supplement preference results for the other tested backbones, or provide a small-scale human evaluation with CTI analysts to validate the LLM judge's conclusions?

**Limitations:**

yes

**Strengths And Weaknesses:**

## Strengths
1. The paper addresses the core challenge of reward sparsity in RLVR for structured CTI generation, with rigorous empirical support across 4 diverse open-weight LLM backbones (3B to 8B parameter scales) and 12 established CTI benchmarks. The proposed MinervaRL delivers consistent performance gains over both base models and pure GRPO-based RLVR baselines, with comprehensive ablation studies (Table 4) verifying the contribution of each core component. The authors also provide formal theoretical analysis in Appendix F to explain how MinervaRL expands empirical support and crosses the detectability threshold for rare correct outputs, strengthening the technical rigor of the work.
2. The curated Minerva-CTI dataset fills a key gap in CTI LLM research, providing a unified suite of 16 verifiable CTI tasks with 32,000 training and 1,200 validation instances, sourced from authoritative standards including MITRE ATT&CK, NVD, and Sigma. The authors implement strict train-test leakage mitigation measures (Section 5.1), such as excluding multiple-choice formats from training, temporal splitting of CVE data, and holding out evaluation tasks from training objectives, ensuring the reliability and generalizability of experimental results. This dataset and the paired verifiers provide a reusable infrastructure for future RL-based CTI LLM research.
3. The framework leverages the inherent verifiability of CTI standards to eliminate the need for costly human preference labeling or learned reward models, making it highly scalable for real-world security operations. The authors demonstrate that MinervaRL preserves general LLM capabilities (Table 3) while improving domain-specific performance, and provide a detailed impact statement addressing the dual-use nature of CTI automation, aligning with responsible AI research principles. The paper is also well-structured and clearly presented, with appendices providing exhaustive implementation details, dataset specifications, and reward function definitions to support full reproducibility.

## Weaknesses
1. While the paper provides an ablation of the distillation learning rate scale γ (Table 5), it does not perform sensitivity analysis for other critical hyperparameters of MinervaRL, including the distillation interval I, per-interval distillation batch cap M, and EMA teacher decay α. The authors note this limitation in Section 8 due to compute constraints, but the absence of this analysis makes it difficult to assess the robustness of the method and provides insufficient guidance for reproducibility and real-world deployment.
2. The paper only compares MinervaRL to a baseline with increased GRPO rollouts (12 vs. 8), but does not benchmark against other established methods for addressing RLVR reward sparsity, such as adaptive exploration, curriculum learning, or off-policy RLVR variants. The related work section (Section 2) also does not provide an in-depth discussion of how MinervaRL differs from and improves upon these existing approaches, weakening the clarity of the paper’s novelty.
3. The paper does not provide a breakdown of performance on long-tail, low-resource tasks in Minerva-CTI (e.g., CAPEC-related tasks with only a few hundred training samples, Table 6), so it is unclear whether MinervaRL delivers consistent gains in data-scarce CTI scenarios. Additionally, the dataset and evaluation benchmarks are exclusively English-centric (noted in Section 8), with no analysis of the method’s performance on multilingual CTI data, limiting the demonstrated generalizability of the framework.
4. The pairwise preference analysis (Section 6.3) only evaluates the Llama-3.1-8B backbone, with no results for the other three tested models, so it cannot confirm that MinervaRL improves response quality across model scales. Furthermore, the analysis relies solely on an LLM judge (GPT-5.2) with no human evaluation to validate the judge’s alignment with real CTI analyst preferences, introducing potential bias and reducing the reliability of the qualitative conclusions.

---

> ### Author Rebuttal · Authors · 2026-03-31
>
> We thank the reviewers for the constructive feedback and address the main points below.
>
> **1. Hyperparameters.** We agree that a sweep over $I$, $M$, and $\alpha$ would strengthen the paper, but these values were chosen deliberately rather than arbitrarily. We prioritized ablating the distillation LR scale $\gamma$ because it most directly controls the strength of the auxiliary self-distillation update relative to RLVR. As Table 5 shows, this knob has a clear trade-off: too small under-utilizes accepted traces, while too large hurts performance.
>
> For $I$ and $M$, our goal was to keep the auxiliary distillation budget bounded relative to the RL budget. Each RL step uses 128 prompts, so one interval of $I=10$ corresponds to 1280 prompts through the RL loop. We then cap distillation at $M=256$ accepted traces, which keeps the SFT phase a modest auxiliary update while also keeping the number of distillation samples roughly stable across training. In our runs, this cap is reached for all models throughout training except Llama-3.1-8B; for that model, the per-interval accepted-trace counts are shown [here](https://tinyurl.com/3zs4zte3). More broadly, we view $I$ and $M$ as coupled knobs rather than fully independent ones: if $I$ increases, $M$ would typically need to be adjusted accordingly.
>
> For the EMA teacher, we use $\alpha=0.995$ as a practical balance between stability and responsiveness: the teacher should be smoother than the online actor when generating ACR traces, but not so stale that it lags behind policy improvement. This is also consistent with standard EMA-teacher practice, where values in the $0.99$--$0.999$ range are common [1].
>
> *Importantly, we use the same $(\gamma,I,M,\alpha)$ settings across all four backbones and still observe consistent gains over GRPO.*
>
> **2. Baselines.** We thank the reviewer for this suggestion. While on-policy approaches such as adaptive exploration and curriculum learning can improve performance empirically, recent work argues that they do not by themselves guarantee expansion beyond the base model’s reasoning boundary under standard on-policy RLVR [2]. In contrast, MinervaRL addresses this bottleneck directly by seeding verified traces for hard prompts and distilling them back onto the original prompt. We support this both formally in Section 4.3 and empirically in Figure 4; additional plots for the other backbones are provided [here](https://tinyurl.com/4jfwj5t7).
>
> Off-policy approaches such as [3] can potentially mitigate reward sparsity by leveraging external SFT traces, whether human-written or generated by a stronger teacher. A key distinction is that MinervaRL does not rely on such external supervision. Instead, it uses self-generated answer-conditioned traces from an EMA teacher of the same model, together with verifier-based filtering and distillation. We believe this is appealing for CTI tasks, where annotated traces may be scarce and stronger domain-specific teacher models may not be available.
>
> Due to computational constraints, we were not able to include additional baselines from adaptive-exploration, curriculum-learning, or off-policy RLVR families in the current draft. However, following Reviewer 6ehu’s suggestion, we added two rejection fine-tuning (RFT) baselines, and these results further support the advantage of MinervaRL over both GRPO and RFT; see [here](https://tinyurl.com/yc57vt8k).
>
> **3. Preference analysis.** We agree that the original qualitative analysis was limited to Llama-3.1-8B and relied only on an LLM judge. To address this, we now add (i) GPT-5.2 pairwise preference evaluation for **all four backbones**, including Base, GRPO, MinervaRL, and the two RFT baselines, and (ii) a small-scale human-validation study.
>
> For human validation, two annotators scored a random set of 100 questions covering the five Llama-3.1-8B variants, and we additionally compared against Claude Sonnet 4.6 on the same set. The rubric and annotation interface are available [here](https://tinyurl.com/5f37djfe), and the full GPT-5.2 pairwise preference results are [here](https://tinyurl.com/39a5nejt). These results show that MinervaRL is preferred over GRPO on all four backbones.
>
> To assess judge reliability, we measured Quadratic Weighted Kappa (QWK) and Spearman correlation on the 100-example annotated set:
>
> | Pair | QWK | Spearman |
> |---|---:|---:|
> | Annotator 1 vs Annotator 2 | 0.6514 | 0.6461 |
> | Annotator 1 vs GPT | 0.6019 | 0.7822 |
> | Annotator 2 vs GPT | 0.5680 | 0.6440 |
> | Annotator Avg vs GPT | 0.6313 | 0.7722 |
> | Claude vs GPT | 0.7634 | 0.8266 |
>
> Ref:
>
> [1] Tarvainen and Valpola, *Mean teachers are better role models: Weight-averaged consistency targets improve semi-supervised deep learning results*, NeurIPS 2017.
> [2] Yue, Yang, et al. "Does reinforcement learning really incentivize reasoning capacity in LLMs beyond the base model?." NeurIPS 2025.
> [3] Yan, Jianhao, et al. "Learning to reason under off-policy guidance." NeurIPS 2025.

---

> > ### Author Rebuttal · Reviewer_QS8q · 2026-04-01
> >
> > Thank you for the detailed rebuttal. While the authors added RFT baselines and discussed differences with other methods, direct empirical benchmarking against adaptive exploration, curriculum learning, or off-policy RLVR (core reward sparsity mitigation approaches) is still missing, and this gap cannot be resolved without additional large-scale experiments. Thus, I will maintain my current score.

---

> > > ### Author Response · Authors · 2026-04-07
> > >
> > > We thank the reviewer for following up on this point. To address the missing empirical comparison directly, we now include **LUFFY** [1], a state-of-the-art off-policy RLVR method, as a baseline in addition to the previously added RFT baselines. For LUFFY, we use the DART dataset we generated, reduced to one accepted trace per training prompt. We ran LUFFY on the same four backbones. The results across all 12 evaluation tasks are below.
> > >
> > > # Llama-3.1-8B
> > >
> > > | Method | CKT | CyberMetric | SOCEval | RCM | VSP | ATE | RMS | ElasticRule | APTNER | LANCE | AnnoCTR | AZERG | Avg. |
> > > |:--|--:|--:|--:|--:|--:|--:|--:|--:|--:|--:|--:|--:|--:|
> > > | STaR-CTI | 70.0 | 81.8 | 61.7 | 57.4 | 73.8 | 29.2 | 9.6 | 21.3 | 33.1 | 80.0 | 46.5 | 32.1 | 49.7 |
> > > | DART-CTI | 71.3 | 82.4 | 61.5 | 64.0 | 73.0 | 37.8 | 27.7 | 31.0 | 34.0 | 74.8 | 47.1 | 39.3 | 53.7 |
> > > | GRPO | 71.9 | *85.4* | 63.0 | 66.3 | 82.6 | 32.0 | 30.9 | 32.2 | 32.7 | *86.2* | 49.5 | 39.5 | 56.0 |
> > > | LUFFY-CTI | 73.9 | 83.7 | 63.3 | 63.8 | 79.6 | 40.6 | 34.3 | 33.8 | *35.4* | 82.4 | 49.7 | 43.1 | 57.0 |
> > > | MinervaRL | *73.9* | 84.2 | *64.7* | *68.8* | *87.6* | *48.4* | *42.1* | *40.5* | 34.1 | 84.6 | *50.3* | *43.7* | *60.2* |
> > >
> > > # Llama-3.2-3B
> > >
> > > | Method | CKT | CyberMetric | SOCEval | RCM | VSP | ATE | RMS | ElasticRule | APTNER | LANCE | AnnoCTR | AZERG | Avg. |
> > > |:--|--:|--:|--:|--:|--:|--:|--:|--:|--:|--:|--:|--:|--:|
> > > | STaR-CTI | 64.3 | 70.5 | 50.3 | 36.4 | 53.4 | 4.2 | 7.0 | 1.9 | 21.4 | 76.3 | 33.8 | 28.0 | 37.3 |
> > > | DART-CTI | *73.0* | 78.2 | 54.5 | 54.9 | 61.2 | 16.8 | 17.8 | 11.6 | 22.5 | 73.8 | 38.7 | 25.7 | 44.0 |
> > > | GRPO | 71.2 | 77.8 | *58.3* | 48.9 | 56.5 | 5.2 | 15.4 | 5.3 | 27.5 | 70.6 | 38.5 | 29.7 | 42.1 |
> > > | LUFFY-CTI | 70.6 | *78.5* | 55.4 | 43.8 | 71.6 | 19.4 | 24.2 | *20.4* | 15.0 | 68.0 | *46.0* | 32.1 | 45.4 |
> > > | MinervaRL | 71.0 | 78.1 | 54.6 | *57.2* | *77.1* | *21.8* | *29.3* | 20.1 | 16.7 | *82.2* | 37.9 | *33.7* | *48.3* |
> > >
> > > # Qwen3-8B
> > >
> > > | Method | CKT | CyberMetric | SOCEval | RCM | VSP | ATE | RMS | ElasticRule | APTNER | LANCE | AnnoCTR | AZERG | Avg. |
> > > |:--|--:|--:|--:|--:|--:|--:|--:|--:|--:|--:|--:|--:|--:|
> > > | STaR-CTI | 37.9 | 49.3 | 65.1 | 36.8 | 68.7 | 15.6 | 3.5 | 4.9 | *37.9* | 55.7 | 40.0 | 26.4 | 36.8 |
> > > | DART-CTI | 39.2 | 55.7 | 65.2 | 47.1 | 72.7 | 14.0 | 6.5 | 16.4 | *37.9* | *76.4* | *46.5* | *36.6* | 42.9 |
> > > | GRPO | 69.8 | 72.7 | *69.0* | 60.5 | 68.8 | 23.6 | 8.2 | 18.1 | 37.7 | 66.1 | 37.8 | 31.1 | 47.0 |
> > > | LUFFY-CTI | *78.6* | *88.3* | 64.8 | *65.3* | 73.2 | 29.4 | *25.8* | *32.4* | 34.3 | 74.5 | 45.1 | 26.8 | 53.2 |
> > > | MinervaRL | 77.8 | 88.2 | 67.5 | 64.8 | *79.4* | *32.0* | 20.1 | 22.0 | 37.4 | 74.9 | 43.0 | 33.1 | *53.4* |
> > >
> > > # Qwen3-4B
> > >
> > > | Method | CKT | CyberMetric | SOCEval | RCM | VSP | ATE | RMS | ElasticRule | APTNER | LANCE | AnnoCTR | AZERG | Avg. |
> > > |:--|--:|--:|--:|--:|--:|--:|--:|--:|--:|--:|--:|--:|--:|
> > > | STaR-CTI | *76.1* | *88.2* | *66.1* | 21.1 | 80.2 | 5.8 | 6.6 | 12.3 | 31.8 | *62.0* | 46.9 | *47.7* | 45.4 |
> > > | DART-CTI | 45.4 | 42.1 | 60.0 | 54.6 | 73.8 | 20.6 | 5.8 | 17.1 | 27.6 | 47.5 | 43.2 | 35.1 | 39.4 |
> > > | GRPO | 74.2 | 85.8 | 63.9 | *60.8* | *88.1* | 20.2 | 3.8 | 20.4 | *32.4* | 58.2 | 39.6 | 24.0 | 47.6 |
> > > | LUFFY-CTI | 54.1 | 56.5 | 60.9 | 59.9 | 72.6 | 17.6 | *10.3* | 23.4 | 29.2 | 60.4 | 46.7 | 36.2 | 44.0 |
> > > | MinervaRL | 70.0 | 80.0 | 64.0 | 59.9 | 80.0 | *25.4* | 5.1 | *27.5* | 28.0 | 56.7 | *50.4* | 29.1 | *48.0* |
> > >
> > > While LUFFY outperforms GRPO on 3 models, MinervaRL remains best on all four backbones by average score. It outperforms LUFFY by `+3.2` on Llama-3.1-8B, `+2.9` on Llama-3.2-3B, and `+4.0` on Qwen3-4B, while being nearly tied on Qwen3-8B.
> > >
> > > A practical distinction is that LUFFY requires a precomputed off-policy trace dataset, obtained from human-annotated SFT traces or a stronger teacher via rejection sampling. In our setup, building the 32k-prompt LUFFY dataset required 1,017,847 teacher attempts or `31.8` attempts per prompt on average before RL training begins. Under our default MinervaRL schedule, each prompt is seen about 2 times on average, with 8 on-policy rollouts per visit and up to 4 additional answer-conditioned generations only for hard prompts, giving an upper bound of `24` generations per prompt across the full RL process. Thus, LUFFY’s offline trace construction alone already requires more attempts per prompt than the full MinervaRL training loop.
> > >
> > > Taken together, these results strengthen our claim that MinervaRL outperforms plain GRPO and fixed-trace RFT baselines, and is competitive with or better than a recent off-policy RLVR method while avoiding a large precomputed trace corpus from a stronger teacher. While adaptive exploration and curriculum learning are also relevant; we view them as allocation strategies for rollout or training effort, whereas MinervaRL improves the learning signal on hard prompts under limited sampling. We will clarify this positioning in the revision.
> > >
> > > Ref
> > >
> > > [1] Yan, Jianhao, et al. “Learning to reason under off-policy guidance.” NeurIPS 2025.

---

### Official Review · Reviewer_6ehu · 2026-03-13

**Soundness:** 3
**Presentation:** 3
**Significance:** 3
**Originality:** 2
**Overall Recommendation:** 4
**Confidence:** 5

**Summary:**

This paper introduces Minerva, a framework for training CTI-specialized LLMs via RLVR. The core insight is that CTI standards define canonical identifiers enabling deterministic verification, making CTI a natural fit for verifiable-reward training. The authors curate a 16-task dataset and propose MinervaRL to address reward sparsity: when a model fails to produce any correct rollout, it conditions generation on the gold label to elicit verified reasoning traces, filters them, and distills them back onto the original label-free prompt.

**Compliance With Llm Reviewing Policy:**

Affirmed.

**Final Justification:**

My all concerns have been addressed by the rebuttal. So I raise my score to 4.

**Key Questions For Authors:**

**Q1.** Could the author add the baselines mentioned in the Weaknesses?

**Q2.** Is reward sparsity concentrated in specific identifier families (e.g., CWE long-tail vs. ATT&CK technique IDs), or roughly uniform across tasks? This is an interesting question, as the answer would reveal which identifier namespaces are better represented in the backbone models' pretraining data — and consequently, which CTI task categories stand to benefit most from MinervaRL's hardness-gated distillation.

**Q3.** Why is CVSS scoring excluded from the ACR hard-example criterion?

**Q4.** The ~37% training overhead is reported as a total figure. Could the authors break down wall-clock time by component — GRPO rollout, reward computation, ACR generation, SFT step — across a representative backbone? The key question is whether MinervaRL is Pareto-dominant over vanilla GRPO at equal compute budget, or whether it requires a strictly larger budget to achieve its gains.

**Q5.** The pairwise evaluation uses GPT-5.2 as the sole judge. Could the authors release their evaluation prompt template?

**Limitations:**

yes

**Strengths And Weaknesses:**

## W1. Incomplete related work.
Several closely related works are not discussed.
- STaR [1] is the foundational "bootstrap reasoning from gold labels" paradigm: it conditions the model on a gold answer to elicit a reasoning trace, filters verified traces, and distills back onto label-free prompts — the same three-step logic as MinervaRL's ACR mechanism.
- DART-Math [2] addresses the identical hard-example coverage problem as MinervaRL's hardness gate: it allocates more sampling trials to queries the model fails on, rather than abandoning them.
- AdaBack [3] dynamically reveals partial prefixes of gold solutions to zero-reward prompts, allowing the model to complete the reasoning chain.
- VulnLLM-R [4] applies a structurally analogous training recipe — gold-label-conditioned reasoning data generation, filtering, and correction — to vulnerability detection, another cybersecurity task with verifiable outputs.

## W2. Missing baseline: iterative rejection sampling fine-tuning.
MinervaRL's performance is compared against plain GRPO and a "+12 rollouts" variant, which rules out the "more sampling" explanation. However, the most structurally similar baseline — iterative rejection sampling fine-tuning (RFT), as in STaR or DART-Math — is absent. In RFT, the model is repeatedly prompted with gold-label-conditioned inputs, verified traces are collected, and the model is fine-tuned on them; this is precisely the ACR+distillation loop without the interleaved GRPO component. Without this baseline, it is unclear whether the gains of MinervaRL over GRPO stem from the distillation component alone, the GRPO component alone, or their interleaving. The ablation in Table 3 does not isolate this.

## W3. Hyperparameter sensitivity undercharacterized.
MinervaRL introduces two key hyperparameters beyond standard GRPO: the hardness threshold γ (zero-reward rollout fraction triggering ACR) and the distillation period α (frequency of SFT interleaving). The paper ablates γ in Table 4 but does not sweep α or the ACR prompt temperature. More importantly, the paper reports a ~37% training overhead from distillation but does not provide wall-clock breakdowns by component or GPU-hours, making it difficult to assess whether MinervaRL is compute-competitive with running vanilla GRPO longer or with more rollouts. A Pareto comparison of performance vs. compute across methods would significantly strengthen the empirical claims.

## W4. Pairwise evaluation relies on a single proprietary judge without a rubric.
The open-ended pairwise preference evaluation uses a single proprietary judge (GPT-5.2) with no published evaluation prompt, rubric, or inter-annotator agreement measure. The author should consider release these information.

[1] Zelikman et al. "STaR: Bootstrapping Reasoning with Reasoning." NeurIPS 2022. https://arxiv.org/abs/2203.14465

[2] Tong et al. "DART-Math: Difficulty-Aware Rejection Tuning for Mathematical Problem-Solving." NeurIPS 2024. https://arxiv.org/abs/2407.13690

[3] Amani et al. "RL for Reasoning by Adaptively Revealing Rationales." arXiv:2506.18110, 2025. https://arxiv.org/abs/2506.18110

[4] Nie et al. "VulnLLM-R: Specialized Reasoning LLM with Agent Scaffold for Vulnerability Detection." arXiv:2512.07533, 2025. https://arxiv.org/abs/2512.07533

---

> ### Author Rebuttal · Authors · 2026-03-31
>
> We thank the reviewers for the constructive feedback and address the main points below.
>
> **1. RFT baselines.** Following the reviewer’s suggestion, we added two iterative rejection fine-tuning baselines for all four backbones, based on **STaR** and **DART-Math**.
>
> For STaR, we run an iterative verifier-filtered bootstrapping loop on the same Minerva train split. In each round, we sample one trace from the original prompt and one gold-answer-conditioned rationalization trace, score both with the Minerva verifier, keep the original trace if verifier-correct, and otherwise keep the rationalization trace only when the original fails and it is verifier-correct. The accepted traces from that round alone are then used to train a fresh SFT model for the next round.
>
> For DART, we build a fixed two-trace-per-question corpus over the same 32k training prompts using a staged process. We first sample only from the original prompt, allowing up to 32 attempts per question and stopping early once two verifier-accepted traces are found. We then fill missing slots with answer-guided generation using the same ACR-style prompting as Minerva, first with the full verifier plus the same heuristic/ML filtering as the RL loop, then with verifier-only acceptance for harder tail cases. Finally, the remaining 29 missing slots are deterministically filled with boxed gold answers so that the final dataset has exactly two traces per question.
>
> The results are [here](https://tinyurl.com/yc57vt8k). MinervaRL consistently outperforms both GRPO and these RFT baselines, which helps separate the sources of gain: the improvement is not explained by distillation alone, nor by RLVR alone, but by their interleaving.
>
> **2. Reward sparsity.** Reward sparsity is not uniform across tasks. Using the DART stage-1 attempts, we find that ATE (scenario → ATT\&CK technique) is clearly sparser than RCM (CVE → CWE): ATE needs more attempts on average (26.7 vs. 22.5), yields fewer verifier-accepted traces (0.66 vs. 0.96), and hits the 32-attempt cap more often (73.2% vs. 56.9%). The plots are [here](https://tinyurl.com/4r5mnfdc). The spread within each panel also shows identifier-level long tails, so the evidence points to both cross-task and within-namespace sparsity.
>
> **3. CVSS exclusion from ACR.** CVSS differs from the other tasks because its reward is based on distance between computed CVSS v3.1 base scores, rather than exact recovery of a long-tail identifier. Each example decomposes into 8 base metrics with small finite option sets, so partial correctness already receives informative reward. As a result, CVSS does not exhibit the same zero-reward sparsity pattern that motivates ACR for long-tail identifier tasks such as ATT\&CK or CWE. In practice, GRPO already performs strongly on CVSS, so applying the same ACR mechanism there is less well motivated.
>
> **4. Training overhead and compute competitiveness.** To isolate overhead, we ran a small-scale timing study on Llama-3.1-8B-Instruct for 10 training steps with validation disabled and one ACR/SFT interleave at step 10. Including end-to-end process time, GRPO took 469.8s and MinervaRL took 654.5s, a 39.3% increase. The MinervaRL breakdown was: GRPO rollout 110.3s, GRPO reward 2.3s, ACR prompt build 2.9s, ACR generation 36.4s, ACR reward 5.9s, and SFT distillation 9.3s. The remaining gap comes mainly from ACR log-prob computation (32.1s) and a modest increase in the shared actor update (+16.9s vs. GRPO). MinervaRL is therefore not strictly Pareto-dominant at equal step count, but the validation curves across all four models are informative: it often surpasses the best GRPO validation score much earlier in training. These curves are [here](https://tinyurl.com/4ebtd38j). So while MinervaRL adds per-step overhead, it is often more compute-efficient in terms of reaching stronger validation performance.
>
> **5. GPT-based qualitative evaluation and related work.** We strengthened the qualitative analysis in two ways. First, we now add GPT-5.2 pairwise preference evaluation for all four backbones, including Base, GRPO, MinervaRL, and the two RFT baselines, using a standardized rubric. Second, we add a small-scale human-validation study. The rubric and annotation interface are [here](https://tinyurl.com/5f37djfe), and the full pairwise preference results are [here](https://tinyurl.com/39a5nejt). These results show that MinervaRL is preferred over GRPO on all four backbones.
> For judge validation, two annotators scored a random set of 100 questions covering the five Llama-3.1-8B variants, and we also compared against Claude Sonnet 4.6 on the same set. Agreement with GPT-5.2 is substantial:
>
> | Pair | QWK | Spearman |
> |---|---:|---:|
> | Annotator 1 vs Annotator 2 | 0.6514 | 0.6461 |
> | Annotator 1 vs GPT | 0.6019 | 0.7822 |
> | Annotator 2 vs GPT | 0.5680 | 0.6440 |
> | Annotator Avg vs GPT | 0.6313 | 0.7722 |
> | Claude vs GPT | 0.7634 | 0.8266 |
>
> Finally, we will enrich Section 2 to better cover related works and baselines.

---

> > ### Author Rebuttal · Reviewer_6ehu · 2026-04-01
> >
> > Thanks for the response. My all concerns have been addressed. I have raised my scores accordingly.

---

> > > ### Author Response · Authors · 2026-04-07
> > >
> > > Thank you for the thoughtful follow-up. We appreciate your time and are glad the rebuttal addressed your concerns.

---

### Official Review · Reviewer_nv8w · 2026-03-16

**Soundness:** 3
**Presentation:** 2
**Significance:** 2
**Originality:** 2
**Overall Recommendation:** 3
**Confidence:** 5

**Summary:**

Cyber Threat Intelligence (CTI) analysts are tasked with transforming unstructured security data into standardized, machine-readable formats for automated processing. While large language models (LLMs) show potential for this task, existing methods struggle with the stability of generating structured CTI outputs and heavily rely on supervised fine-tuning (SFT). This paper proposes a novel approach by leveraging the inherent structure of CTI data and community-maintained resources, which define standardized identifiers and patterns, to create a reinforcement learning with verifiable rewards (RLVR) method. We present **Minerva**, a unified framework that addresses multiple CTI sub-tasks and integrates task-specific verifiers that provide scores for structured outputs and identifier predictions. To overcome the reward sparsity problem often encountered during model sampling, we design a lightweight self-training mechanism that generates additional verified trajectories and distills them back into the model.

Experiments on various large language model architectures demonstrate that Minerva consistently outperforms traditional supervised fine-tuning approaches, improving model accuracy and robustness across multiple benchmark tests.

**Compliance With Llm Reviewing Policy:**

Affirmed.

**Final Justification:**

Most of the reviewer' concerns have been addressed in the rebuttal stage, and the reviewer will raise their scores accordingly.

**Key Questions For Authors:**

Please refer to the Weaknesses.

**Limitations:**

yes

**Strengths And Weaknesses:**

Strengths:

1. Practical Relevance and Clear Problem Definition: The work focuses on a very practical issue within Cyber Threat Intelligence (CTI): converting unstructured security text into standardized, machine-processable structured outputs. This is not a vague NLP task but directly addresses a real pain point in security analysis workflows.

2. Task Naturally Suited for Verifiable Learning: CTI tasks involve many standardized identifiers, fixed schemas, set labels, and outputs that can be programmatically checked, making “verifier-driven training” inherently reasonable.

3. Complete and Cohesive Methodology: This work not only proposes RLVR but also incorporates answer-conditioned reasoning, trajectory filtering, and periodic distillation, forming a complete “sampling → validation → filtering → distillation” training loop. This closed loop directly addresses the reward sparsity issue in a targeted manner, rather than simply stacking modules.

4. Tackling Long-Tail & Sparse Reward Challenges: The work is not just focused on easy tasks but targets the long-tail labels, rare mappings, and difficulty in sampling correct trajectories, which are typical challenges in CTI tasks.

5. Unified Dataset and Training Framework: The Minerva-CTI dataset consolidates multiple CTI sub-tasks into one training suite, facilitating a unified evaluation and training process.

Weaknesses:

1. Incremental Contribution Rather Than a Novel Paradigm: Components like GRPO, RLVR, self-training, distillation, rationale generation, and filtering are not new concepts. The reviewer feels this work is an incremental effort that combines existing methods into the CTI domain, rather than introducing a fundamentally new learning paradigm.

2. Unclear Definition of "Structured Threat Intelligence": The definition of “structured threat intelligence” is too broad. It includes identifier prediction, multi-label sets, structured field generation, entity-relationship extraction, etc. For readers who are completely unfamiliar with the topic, the paper might benefit from a brief, concrete example in the introduction to clarify this concept and lower the entry threshold.

3. RLVR Extension Mechanism Seen as Self-training with Verifier or Pseudo-label Distillation: The reviewer believes the “RLVR extension mechanism” described in the paper is essentially a self-training or pseudo-label distillation process with a verifier, rather than a fundamentally different reinforcement learning mechanism.

4. Many "Effective but Engineering-Focused" Components: There are many practical components in this work, such as filters, caching, sampling strategies, and threshold settings. While these are certainly important, the reviewer feels that unless the paper clearly distinguishes which parts are core innovations and which are mere implementation details, the theoretical novelty of the paper may seem less advanced.

5. Ablation Study Does Not Fully Demonstrate “What Is Driving the Effectiveness”: The ablation study does not conclusively show whether the improvements are driven by ACR, filtering, distillation, or due to additional label information and increased computation.

6. Possible Label Leakage in Constructing Answer-Conditioned Trajectories: The paper constructs answer-conditioned trajectories using real labels in training. This could be seen as label leakage or additional supervision. If this is not the case, the paper should clarify how inference avoids using the real labels and how the training stage ensures a fair comparison.

7. Excessive Conceptual Packaging with Insufficient Precision: Some terms like “experience support ability,” “zero-reward prompts,” and “difficulty-gated answer-conditioned reasoning” seem to overemphasize conceptual drama rather than enhancing technical clarity. These terms, though conceptually understandable, lack precision in definition, which might lead readers to think that the language is more theatrical than informative.

8. Increased Training Costs Need Better Cost-Benefit Justification: While the increased training cost is acknowledged, the paper should provide a better explanation of its cost-effectiveness. For practical applications, training costs are a significant burden. Is the performance gain from this complex multi-stage training worth the additional computational expense? How does this compare to simpler approaches like “increased sampling budget” or “finer-grained SFT”? Is the advantage significant enough?

9. Verifiers' Design Logic: The reviewer is more interested in understanding the logic behind the design of the verifiers—why this specific design accurately reflects the quality of CTI tasks. Are there cases where the verifiers cannot capture “semantically correct but format errors”?

---

> ### Author Rebuttal · Authors · 2026-03-31
>
> We thank the reviewers for the constructive feedback and address the main points below.
>
> **1. Novelty.** We agree that MinervaRL combines existing ingredients, namely on-policy RLVR with periodic answer-conditioned self-distillation. Our novelty claim is not that each ingredient is new in isolation, but that their combination directly addresses a CTI-specific bottleneck. Under limited-rollout RLVR, many hard CTI prompts yield no successful rollout because the target spaces are long-tailed and verifier constraints are strict. MinervaRL addresses this by using answer-conditioned generation only for such hard prompts, filtering verified high-quality traces, and distilling them back onto the original answer-free prompt. We support this both theoretically in Section 4.3 / Appendix F and empirically in Figure 4, where MinervaRL reduces the zero-solve fraction beyond what is achieved by simply increasing GRPO rollouts. Additional plots for all four backbones are provided [here](https://tinyurl.com/4jfwj5t7). Finally, to the best of our knowledge, this is the first CTI-focused RLVR framework built around verifier-checkable CTI tasks, rather than a direct transfer of RLVR recipes from math/code.
>
> **2. What drives the gains.** We also took additional steps to separate the sources of gain. Table 4 already compares against GRPO and GRPO with increased rollouts (12 vs. 8), showing that the effect is not explained by extra sampling alone. We now also add two rejection fine-tuning (RFT) baselines, which introduce extra supervision/self-training without the full MinervaRL loop. MinervaRL consistently outperforms both GRPO and these RFT baselines, suggesting that the gains are not explained by RLVR alone, nor by added supervision alone, but by their combination: RLVR provides the verifier-driven optimization signal, while targeted answer-conditioned distillation improves learning on sparse-reward hard prompts. Results for the added baselines are [here](https://tinyurl.com/yc57vt8k).
>
> **3. Additional supervision, label leakage, and fair comparison.** While answer-conditioned trajectories do use the gold label during training, this is not test-time leakage: at inference and benchmark evaluation, all methods use exactly the same original answer-free prompts. The gold label is used only to construct auxiliary self-distillation targets during training. More broadly, all methods have access to the same labeled training data; MinervaRL differs only in how it uses those labels to improve learning on hard prompts. This makes the comparison fair with GRPO, RFT, and SFT-style baselines.
>
> **4. Structured CTI outputs and terminology.** By “structured threat intelligence,” we mean outputs aligned to standardized CTI schemas and identifiers, such as ATT&CK technique/tactic/mitigation IDs, CWE/CVSS prediction, and STIX-style entity/relation extraction. A short concrete example in the introduction would make this clearer for readers less familiar with CTI, and we will add one. We will also make the separation between core ideas and implementation details more explicit. The core method is the hard-prompt ACR generation + verified filtering + periodic distillation loop; items such as caching, thresholds, and similar choices are implementation details. On terminology, we do not use the word “experience” in the paper. If the reviewer is referring to “empirical support,” that term follows prior RLVR literature rather than being introduced by us. We will simplify wording where helpful.
>
> **5. Filtering.** The paper provides both methodological details and empirical evidence for filtering. Appendix C describes the filtering pipeline, which removes leakage and low-quality traces before distillation, and Table 4 shows that removing filtering reduces performance relative to full MinervaRL. More broadly, filtering is not intended as the core innovation by itself, but as an important stabilizing component that improves the quality of the traces used for self-distillation.
>
> **6. Verifier design.** Our tasks are built around canonical CTI standards, so the verifiers are intentionally tied to standardized identifiers and structured outputs. At evaluation we already use more permissive answer extraction, which helps recover semantically correct answers despite minor formatting variation; remaining noise can still arise from imperfections in public-source annotations. We will clarify this design in the revision.
>
> **7. Cost-benefit.** We already compare against increased sampling budget (GRPO-12) and strong security SFT baselines (Table 1: Llama-Primus-Merged, Foundation-Sec-8B-Instruct), and MinervaRL outperforms both. We also provide validation curves across all four models [here](https://tinyurl.com/4ebtd38j), where MinervaRL often surpasses the best GRPO validation score substantially earlier in training, often around 130-220 steps versus the best GRPO score appearing much later (400+). This helps justify the added training complexity in practice.

---

> > ### Author Rebuttal · Reviewer_nv8w · 2026-04-03
> >
> > Most of the reviewer' concerns have been addressed in the rebuttal stage, and the reviewer will raise their scores accordingly.

---

> > > ### Author Response · Authors · 2026-04-07
> > >
> > > Thank you for the thoughtful follow-up. We appreciate your time and consideration.

---

### Decision · Program_Chairs · 2026-04-30

**Decision:**

Reject

**Comment:**

Thanks for your submission to ICML 2026. This submission proposes Minerva, a dataset and training pipeline for RLVR for Cyber Threat Intelligence (CTI) tasks. Authors and all reviewers were actively engaged in the discussion phase. After discussion, the scores are all borderline and diverge. Reviewers agree with the strength of the paper in enabling RLVR for CTI tasks and achieving strong performance. Specifically, Minerva solves practical challenges such as reward sparsity and warm-up start hardness, and techniques are generalizable to other domains as results shown during the discussion phase. However, on the other hands, reviewers still share concerns on the degree of contribution beyond integrading existing RLVR techniques; and relatively limited evaluation to isolate the contribution of each component and compare with corresponding similar techniques in the literature for each component. Overall, I believe that the Minerva framework would be of very practical value for cybersecurity practitioners. But the scientific values in the context of RLVR may need to be further strengthened for publication. As a result, I would recommend the authors to refine the manuscript further based on the feedback and submit this impactful work to a near future venue.